# WAN3DNS: Weak Adversarial Networks for Solving 3D Incompressible Navier-Stokes Equations

## Abstract

The 3D incompressible Navier-Stokes (NS) equations model essential fluid phenomena, including turbulence and aerodynamics, but are challenging to solve due to nonlinearity and limited solution regularity. Despite extensive research, the full mathematical understanding of the 3D incompressible Navier-Stokes equations continues to elude scientists, highlighting the depth and difficulty of the problem. Classical solvers are costly, and neural network-based methods typically assume strong solutions, limiting their use in underresolved regimes. We introduce WAN3DNS, a weak-form neural solver that recasts the equations as a minimax optimization problem, allowing learning directly from weak solutions. Using the weak formulation, WAN3DNS circumvents the stringent differentiability requirements of classical physics-informed neural networks (PINNs) and accommodates scenarios where weak solutions exist, but strong solutions may not. We evaluated WAN3DNS's accuracy and effectiveness in three benchmark cases: the 2D Kovasznay, 3D Beltrami, and 3D lid-driven cavity flows. Furthermore, using Galerkin's theory, we conduct a rigorous error analysis and show that the $L^2$ training error is controllably bounded by the architectural parameters of the network and the norm of residues. This implies that for neural networks with small loss, the corresponding $L^2$ error will also be small. This work bridges the gap between weak solution theory and deep learning, offering a robust alternative for complex fluid flow simulations with reduced regularity constraints.

## 1 Introduction

Despite extensive research, a full mathematical understanding of the 3D incompressible Navier-Stokes equations continues to elude scientists, highlighting the depth and difficulty of the problem. The application of classical numerical methods such as finite elements and finite differences to solve high-dimensional partial differential equations (PDEs) has been challenging due to the notorious curse of dimensionality Han et al. (2018). Integrating neural networks into the solution of PDEs has revolutionized traditional numerical methodologies, offering data-driven flexibility and scalability Weinan et al. (2021); Dissanayake & Phan-Thien (1994); Lagaris et al. (1998). The existing neural network-based approaches for PDEs can be broadly categorized into three paradigms, each with different theoretical foundations and application scopes Tanyu et al. (2023): PINN-based models Raissi et al. (2019); DeepONet-based models Lu et al. (2019) and WAN-based models Zang et al. (2020). The descriptions of the PINNs-based models and DeepNet-based models are in the appendix A, while this paper focuses primarily on the WAN-based models.

Emerging as a third avenue, weak formulation-based methods, such as weak adversarial networks (WAN) Zang et al. (2020), transform PDEs into integral weak forms, transforming the solving process into a minimax optimization problem akin to generative adversarial networks. Li et al. (2024) applied WAN to solve two-dimensional Navier-Stokes equations. Still, it required a stream function and makes error analysis challenging; hereinafter Li et al. (2024) is referred to as WAN-Biharmonic.

However, its potential in fluid mechanics remains underexplored, particularly for incompressible three-dimensional NS equations. The stream function is a scalar function used to describe the 2D incompressible flow; utilizing the stream function can directly satisfy the incompressibility condition.

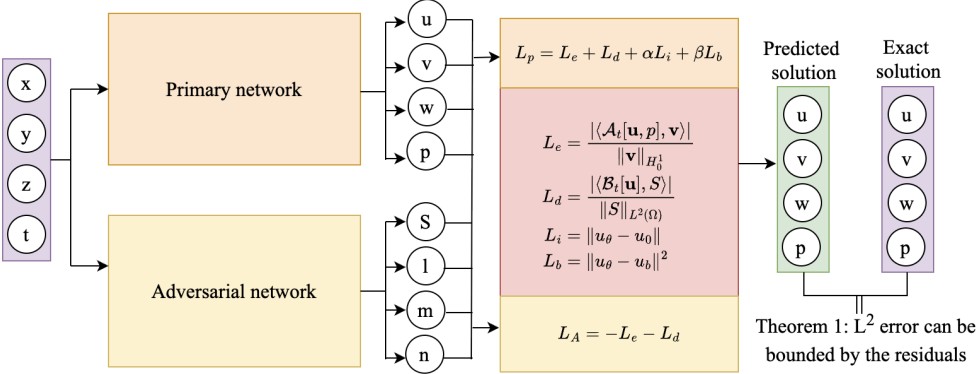

Figure 1: Workflow of WAN3DNS. In the figure, the input $x, y, z, t$ are independent variables, the primary network is a fully connected network for generating the velocity $u, v, w$ and pressure $p$. The adversarial network generates a test function $S$ for divergence-free condition and $l, m, n$, the elements of $\mathbf{v}$, the test function for governing equations. The loss function set according to the equations and conditions finally outputs the solution.

Employing the stream function approach for weak solution computation increases the number of differentiation operations during training. Most industrial problems manifest in three-dimensional forms, rendering the 2D NS equations insufficient for real-world applications. However, the 3D NS equations continue to challenge researchers due to unresolved questions about the existence of global solutions Hoff (1995) and the complexity of their physical phenomena Quartapelle (2013).

Crucially, whereas the original WAN and WAN-Biharmonic are limited to scalar equations, our proposed WAN3DNS method successfully handles vector-valued ones. This advancement is not a trivial matter of increasing network depth; instead, it requires a sophisticated reformulation of the loss function within the weak adversarial framework to inherently enforce the divergence-free constraint and manage the complexities of three-dimensional vector fields. The contributions lie in providing an algorithm to simulate complex flows governed by the 3D NS equations. Specifically:

(I) **Min-max reformulation of the Navier–Stokes equations**: We transfer the 3D incompressible NS equations into a min-max problem and propose **WAN3DNS** a neural algorithm that effectively solves 2D and 3D NS problems within a single framework.

(II) **Theoretical error analysis via Galerkin theory:** We provide a theoretical analysis of the algorithm's approximation error using Galerkin theory, showing that the $L^2$ error of the solution can be bounded by the proposed loss function under certain regularity assumptions. This result directly connects optimization performance to physical accuracy, offering a theoretical foundation for model training.

(III) **Empirical validation on benchmark fluid dynamics problems:** We evaluate the proposed algorithm on three numerical experiments: 2D Kovasznay flow, 3D Beltrami flow, and 3D lid-driven cavity flow, showing that WAN3DNS achieves higher accuracy than DeepXDE, NSFnets and WAN-Biharmonic.

The above method can be summarized as a workflow 1. The remainder of this paper is organized as follows. Section 2 introduces the work related to the application of WAN to the NS equation; Section 3 introduces the proposed WAN3DNS algorithm and convergence analysis. Section 4 covers the three experimental settings considered. Finally, Section 5 analyzes the errors and provides a discussion.

## 2 BACKGROUND AND RELATED WORK

We begin by reviewing the 3D incompressible Navier-Stokes (NS) equations, then describe the distinction between strong and weak PDE formulations. Finally, we introduce Weak Adversarial Networks (WANs), upon which our method is built.

## 2.1 3D Navier-Stokes Equations

Consider the spatial-temporal domain $\Omega \times [0, T]$, where $\Omega \subset \mathbb{R}^3$ is a bounded, simply connected open set. The spatial variable is denoted by $\mathbf{x} = (x, y, z)$, and the velocity $\mathbf{u}(\mathbf{x}, t)$ can be regarded as a solution to the following Navier-Stokes equations:

$$\begin{cases} \frac{\partial \mathbf{u}}{\partial t} + (\mathbf{u} \cdot \nabla)\mathbf{u} = -\nabla p + \nu \Delta \mathbf{u}, & (\mathbf{x}, t) \in \Omega \times [0, T], \\ \nabla \cdot \mathbf{u} = 0, & (\mathbf{x}, t) \in \Omega \times [0, T], \\ \mathbf{u} = \mathbf{u}_\Gamma(t, \mathbf{x}), & \mathbf{x} \in \partial\Omega, t \in [0, T], \\ \mathbf{u} = \mathbf{u}_0(\mathbf{x}), & \mathbf{x} \in \Omega \cup \partial\Omega, t = 0. \end{cases} \quad (1)$$

Here, $p(\mathbf{x}, t)$ is a scalar representing pressure, and $\nu > 0$ denotes the viscosity of the fluid. The boundary velocity $\mathbf{u}_\Gamma$ satisfies the global condition

$$\oint_{\partial\Omega} \mathbf{n} \cdot \mathbf{u}_\Gamma \, dS = 0, \quad \forall t \geq 0, \quad (2)$$

where $\mathbf{n}$ represents the outward normal vector of the boundary $\partial\Omega \times [0, T]$, and $\oint$ denotes the second-type surface integral over a closed loop. The initial velocity field $\mathbf{u}_0$ is divergence-free, i.e., $\nabla \cdot \mathbf{u}_0 = 0$.

Although some theoretical work has been done on error estimation in deep learning for fluid dynamics Biswas et al. (2022), it remains largely confined to 2D settings. In contrast, the convergence behavior of neural networks for solving 3D NS equations is still an open and underexplored challenge in CFD.

## 2.2 Strong Formulation vs Weak Formulation

Two types of solutions are commonly defined for PDEs Evans (2022). The first, known as a *classical solution*, requires that all derivatives appearing in the equation exist and are continuous, and that the solution satisfies the boundary and initial conditions in the classical sense. These solutions were the primary focus of study during the 18th and 19th centuries. The second type, referred to as a *weak* or *generalized solution*, allows for derivatives in a broader sense, accommodating functions that may lack classical differentiability.

The importance of weak solutions arises from two key factors: 1) Certain nonlinear equations do not admit classical solutions, and 2) for some problems, classical solutions cannot be directly obtained. In such cases, one typically constructs a weaker form first and then uses tools from functional analysis to establish sufficient regularity, potentially recovering classical solutions.

## 2.3 Weak Adversarial Networks (WAN)

The WAN algorithm is a neural network-based method for solving PDEs according to their weak form. There are 3 steps in the original WAN Zang et al. (2020) algorithm:

**Step 1: (Reformulate PDEs to optimization problem)** Transform PDEs into weak integral forms. Consider a PDE with initial and boundary conditions, we write in the form

$$L(u) = f, \quad B(u) = g, \quad I(u) = q \quad (3)$$

, where $L$ is a partial differential operator, and $B$ is the boundary operator, and $I$ is the initial operator. WAN rewrite Eq. equation 3 as a minimization in a suitable dual space.

$$u^* = \underset{w \in W}{\mathrm{argmin}}(\|L(u - w)\|_{H^{-1}(\Omega), op} + \alpha\|g - w\|_{L^2(\partial\Omega)} + \beta\|q - w\|_{L^2(\Omega)}) \quad (4)$$

where $\alpha, \beta$ are the penalty coefficients, and

$$\|L(v)\|_{H^{-1}(\Omega), op} = \sup_{\phi \in H_0^1(\Omega), \phi \neq 0} \frac{a(v, \phi)}{\|\phi\|_W}. \quad (5)$$

Here, $a : H^1(\Omega) \times H_0^1(\Omega) \to \mathbb{R}, a(v, \phi) = (L(v), \phi)_\Omega$ is the bilinear form that corresponds to the operator $L$.

**Step 2: (Minimax reformulation of the optimization problem)** Use neural networks to discretize the objective function equation 4, turning the solving process into a minimax optimization problem. WANs build a primary network to approximate $w$, the loss function they build is according to the "$argmin$" in the equation. equation 4. Then build an adversary network to approximate $\phi$, its objective is to make "$sup$" in Eq. equation 5 come true.

**Step 3: (Adversarial Network Training)** Train the network. Choose appropriate parameters for the primary network and the adversary network, arrange them as generative adversarial networks (GAN) Goodfellow et al. (2020), and train the network to obtain the solution.

While the WAN framework is good for equations that have no smooth solution, it is limited to a single governing equation, making it unsuitable for a group of equations. We address this limitation by using WAN3DNS, allowing us to tackle the complex system: the 3D NS equations and enabling simulations of realistic fluid flows. Our method is summarized as Algorithm 1, and important notations are listed in Table 5 in the Appendix.

## 3 WAN3DNS

WAN only solve the scalar equation; they just consider the weak form of one equation. it is difficult to reveal real-world physics in just one dimension. This issue can be addressed by incorporating the number of terms of the loss function. To achieve this, we develop a weak formula for 3D NS equations and adapt it to WAN for simulating flow velocity.

### 3.1 WEAK FORMULATION

The weak solutions of the NS equations, which allow for low regularity and thus better align with physical reality, have garnered considerable attention from researchers. With regard $u$ as the map from $t$ to $\mathbf{x}$, this is a map of the function space $H_0^1(\Omega)$, i.e.

$$\mathbf{u} : [0,T] \rightarrow H_0^1(\Omega), \quad [\mathbf{u}(x)](t) \triangleq \mathbf{u}(x,t), \quad \mathbf{x} \in \Omega, 0 \le t \le T.$$

For fixed vector function $\mathbf{v} \in H_0^1(\Omega)$, and scalar function $S \in L^2(\Omega)$, according to equation 1 we get

$$\left(\frac{\partial \mathbf{u}}{\partial t}, \mathbf{v}\right) + b(\mathbf{u}, \mathbf{u}, \mathbf{v}) + \nu(\nabla \mathbf{u}, \nabla \mathbf{v}) - (p, \nabla \cdot \mathbf{v}) = 0, \qquad \sum_{i=1}^{3}(\frac{\partial S}{\partial x_i}, u_i) = 0 \qquad (6)$$

here $(\cdot, \cdot)$ represents the dual product between $H^{-1}(\Omega)$ and $H_0^1(\Omega)$, $b(\mathbf{u}, \mathbf{v}, \mathbf{w}) = \sum_{i,j=1}^{d}(\mathbf{u}_i \partial_j \mathbf{v}_i, \mathbf{w}_i)$ is a trilinear formula. Based on the weak form of the governing equations and the incompressible condition, we define the operators.

$$\mathcal{A}_t[\mathbf{u}, p] := \frac{\partial \mathbf{u}}{\partial t} - \nu \Delta \mathbf{u} + (\mathbf{u} \cdot \nabla)\mathbf{u} + \nabla p, \qquad \mathcal{B}_t[\mathbf{u}] := \nabla \cdot \mathbf{u}, \qquad (7)$$

where $\mathbf{u} \in H^2(0,T;H^1(\Omega)) \cap L^\infty(0,T;(H^2(\Omega))^d)$ and $p \in D'(\Omega)$. By integration by parts, the dual product $\langle \mathcal{A}_t[\mathbf{u}], \mathbf{v} \rangle$ and $(\mathcal{B}_t[\mathbf{u}], S)$ can be computed as

$$\langle \mathcal{A}_t[\mathbf{u}, p], \mathbf{v} \rangle = \left(\frac{\partial \mathbf{u}}{\partial t}, \mathbf{v}\right) + \nu(\nabla \mathbf{u}, \nabla \mathbf{v}) + b(\mathbf{u}, \mathbf{u}, \mathbf{v}) + (p, \nabla \cdot \mathbf{v}),$$

$$(\mathcal{B}_t[\mathbf{u}], S) = \sum_{i=1}^{d}(\frac{\partial S}{\partial x_i}, u). \qquad (8)$$

The norm of the operator $\mathcal{A}_t[\mathbf{u}, p]$ and $\mathcal{B}_t[\mathbf{u}]$ is defined as

$$\|\mathcal{A}_t[\mathbf{u}, p]\|_{H^{-1}} := \sup_{\mathbf{v} \in (H_0^1(\Omega))^d} \frac{|\langle \mathcal{A}_t[\mathbf{u}, p], \mathbf{v} \rangle|}{\|\mathbf{v}\|_{H_0^1}}, \quad \|\mathcal{B}_t[\mathbf{u}]\|_{H^{-1}} := \sup_{S \in L^2(\Omega)} \frac{|\langle \mathcal{B}_t[\mathbf{u}], S \rangle|}{\|S\|_{L^2(\Omega)}}, \qquad (9)$$

### 3.2 LOSS FUNCTION AND NETWORK ARCHITECTURE

Since $\|\mathcal{A}_t[\mathbf{u}]\|_{H^{-1}} \ge 0$ and $\|\mathcal{B}_t[\mathbf{u}]\|_{H^{-1}} \ge 0$, solving the energy equation and the incompressibility condition is equivalent to finding the solution of the following problems:

$$\min_{\mathbf{u}} \|\mathcal{A}_t[\mathbf{u}]\|_{H^{-1}}^2 = \min_{\mathbf{u}} \max_{\mathbf{v}} \frac{|\langle \mathcal{A}_t[\mathbf{u}, P], \mathbf{v} \rangle|^2}{\|\mathbf{v}\|_{H_0^1}^2}, \qquad (10)$$

$$\min_{\mathbf{u}} \|\mathcal{B}_t[\mathbf{u}]\|_{H^{-1}}^2 = \min_{\mathbf{u}} \max_{S} \frac{|\langle \mathcal{B}_t[\mathbf{u}], S \rangle|^2}{\|S\|_{L^2(\Omega)}^2}. \tag{11}$$

We parameterize $\mathbf{u}$ and $P$ as a neural network $\mathbf{u}_\theta$, $P_\theta$, and parameterize $v, S$ as the outputs of an adversarial network, denoted as $v_\eta, S_\eta$, here $\theta$ and $\eta$ are the hyperparameters of neural networks. In order to minimize the operator norms of $\mathcal{A}_t[u_\theta]$, $\mathcal{B}_t[u_\theta]$, we design a GAN.

According to the interior equation, the incompressible condition, the initial condition, and the boundary conditions. We define the objective function of $u_\theta$ as follows.

$$L_e(\theta, \eta) \triangleq \frac{1}{N_e} \sum_{j=1}^{N_e} \frac{|\langle \mathcal{A}_t[u_\theta], v_\eta \rangle|^2}{\|v_\eta\|^2}(x_j, y_j, z_j), \qquad L_d(\theta, \eta) \triangleq \frac{1}{N_d} \sum_{j=1}^{N_d} \frac{|\langle \mathcal{B}_t[u_\theta], S_\eta \rangle|^2}{\|S_\eta\|^2}(x_j, y_j, z_j),$$

$$L_i(\theta) \triangleq \frac{1}{N_I} \sum_{j=1}^{N_I} \|u_\theta - u_0\|^2(x_j, y_j, z_j), \qquad L_b(\theta) \triangleq \frac{1}{N_b} \sum_{j=1}^{N_b} \|u_\theta - u_b\|^2(x_j, y_j, z_j),$$

$$L_u(\theta, \eta) \triangleq L_e(\theta, \eta) + \alpha L_d(\theta, \eta) + \beta L_i(\theta) + \gamma L_b(\theta). \tag{12}$$

where $L_e$ is the loss function for the governing equation, $L_d$ is for divergence free, $L_i$ initial condition and $L_b$ boundary conditions. $N_e$, $N_d$, $N_I$, $N_b$ denote the numbers of training data for different terms; the weighting coefficients $\alpha, \beta, \gamma > 0$ are hyperparameters as penalty terms.

The test function $v, S$ is only related to spatial variables, and not to time as a variable. We just define the loss function on the adversarial network as:

$$L_v(\theta, \eta) = -L_e(\theta, \eta) - L_d(\theta, \eta) + L_B(\eta),$$

where $L_e(\theta, \eta)$ are the same as the above definition, $L_B(\eta) = \frac{1}{N_b} \sum_{j=1}^{N_b} \|\mathbf{v}_\eta - 0\|^2 + \|\mathbf{S}_\eta - 0\|^2$ is the boundary conditions for the test function because the trace of $\mathbf{v}$ and $S$ are 0. Choosing appropriate hyperparameters, we train the two networks in rotation to find the saddle point of equation 10 equation 11 as the approximation to $u$.

### 3.3 ERROR ANALYSIS

We provide the approximation error analysis to reveal the relation between the residual value the $L^2$ error. All experiments in this paper use Dirichlet boundary conditions, and the analysis is carried out in the Dirichlet setting. First, we write the corresponding pointwise residues as

$$\begin{aligned} R_s(x, t) &= \mathbf{u}_\theta(x, t) - \mathbf{u}_\Gamma(x, t), \quad x \in \partial D, \\ R_{PDE} &= \partial_t \mathbf{u}_\theta + (\mathbf{u}_\theta \cdot \nabla)\mathbf{u}_\theta - \nu \Delta \mathbf{u}_\theta. \end{aligned} \tag{13}$$

The proof of the following Theorem can be seen in Appendix C.

**Theorem 1** *Let $\mathbf{u} \in H^1$, $\mathbf{u} \in H^2(0, T; H^1(\Omega)) \cap L^\infty(0, T; H^2(\Omega))$ be a weak solution of the Navier-Stokes equations. Let $\mathbf{u}_\theta, \mathbf{v}_\theta, q_\theta$ be functions constructed by the set of network parameters $\theta, \phi$, which are the output of the WAN3DNS algorithm. Then, the $L^2$ error of the result can be bounded as follows:*

$$\int_D \|u(x, t) - u_\theta(x, t)\|_2^2 dx dt \leqslant C_2(1 + C_3 T e^{C_3 T}), \tag{14}$$

*where:*

$$\begin{aligned} C_1 =& C(\|u\|_{C^1}, \|\hat{u}\|^{C^1}), \\ C_2 =& \|R_t\|_{L^2(D)}^2 + C_1\sqrt{T|D|}\|R_{div}\|_{L^2(D)} \\ &+ C_1(1 + \nu)\sqrt{T|\partial D|}\|R_s\|_{L^2(\partial D \times [0,T])} + \|R_{PDE}\|_{L^2}^2, \\ C_3 =& 2d^2\|\nabla u\|_{L^\infty(D)} + 1. \end{aligned} \tag{15}$$

Table 1: Relative $L^2$ error and training time comparison for Kovasznay flow. For both u and v, our accuracy is nearly 5 times higher than that of WAN-Biharmonic.

| | $u_{error}$ | $v_error$ | $\psi_{error}$ | $P_{error}$ | Training time (s) |
|---|---|---|---|---|---|
| WAN-Biharmonic | 0.023 | 0.278 | 0.019 | - | 1,213.614 |
| DeepXDE | 0.003 | 0.016 | - | **0.006** | **680.947** |
| NSFnets | 0.064 | 0.219 | - | 3.760 | 727.386 |
| **WAN3DNS** | **0.002** | **0.012** | - | 0.008 | 682.607 |

**Remark 1** *We would like to note that one can also prove a stability result for $\|P - P_\theta\|_{L^2(\Omega)}$ in a similar spirit to Theorem 1. The main steps consist of taking the divergence of the NS equations, using the identity*

$$-\Delta P = Trace((\nabla \mathbf{u})^2) = \sum_{i,j} u^i_{x_j} u^j_{x_i}$$

*and rewriting the result in terms of the different residuals.*

## 4 NUMERICAL VALIDATION

We evaluate the performance of WAN3DNS on 3 benchmark problems of increasing complexity: 1) a classical analytical solution to the incompressible Navier-Stokes equations used to validate accuracy in the 2D setting (Kovasznay flow), 2) A smooth unsteady 3D flow with an exact solution (Beltrami flow). 3) A 3D recirculating flow of a Newtonian fluid inside a cube generated by the shear from a moving lid (Lid-driven cavity flow). The property and number of sampling points for all these 3 examples are reported in Table 6 in the appendix. We compare the accuracy and training time of the WAN3DNS to DeepXDE, NSFnest, WAN-Biharmonic and PINN, which are the current state-of-the-art models. All these models are implemented using the same hyperparameter setting, listed in Table 4 in the Appendix. The presentation of the baseline models is provided in Appendix D.

### 4.1 DOES WAN3DNS WORK ON 2D PROBLEMS?

Kovasznay flow Kovasznay (1948) is a fundamental benchmark case because it provides a rare, non-trivial analytical solution to the steady-state Navier-Stokes equations, making it ideal for validating the accuracy of new computational fluid dynamics solvers. we use Kovasznay flow on $[-0.5, 1.5] \times [-0.5, 1.5]$ as the first example to evaluate WAN3DNS and other 3 baseline model. Kavasznay is a steady flow, so there is no initial condition. The boundary conditions of Kovasznay:

$$\begin{cases} u_1(-0.5, y) = 1 - e^{-0.5\xi} \cos(2\pi y), \\ u_2(-0.5, y) = \frac{\xi}{2\pi} e^{-0.5\xi} \sin(2\pi y), \\ u_1(1.5, y) = 1 - e^{1.5\xi} \cos(2\pi y), \\ u_2(1.5, y) = \frac{\xi}{2\pi} e^{1.5\xi} \sin(2\pi y), \\ u_1(x, -0.5) = u_1(x, 1.5) = 1 + e^{\xi x}, \\ u_2(x, -0.5) = u_2(x, 1.5) = 0. \end{cases} \tag{16}$$

where

$$\xi = \frac{1}{2\nu} - \sqrt{\frac{1}{4\nu^2} + 4\pi^2}, \qquad \nu = \frac{1}{Re} = \frac{1}{40}.$$

The experimental setup can be seen in Table 6 of the appendix. We also list the results obtained when the network size is different in Table 7 in the appendix, which show that WAN performs best, although the parameter setting changes. By eliminating the pressure term $P$ through weak enforcement of incompressibility constraints, our method focuses on the prediction of velocity. We compare WAN3DNS to the other 3 baseline models in terms of relative $L^2$-errors (Table 1) and provide a visual comparison (Figure 2) of $u$; the comparison of $v$ and $p$ is shown in Figure 4 and Figure 5 of the Appendix.

Both pointwise errors and $L^2$ norms demonstrate our superior accuracy over the other 3 models, and DeepXDE is slightly less accurate but more efficient (Table 1).

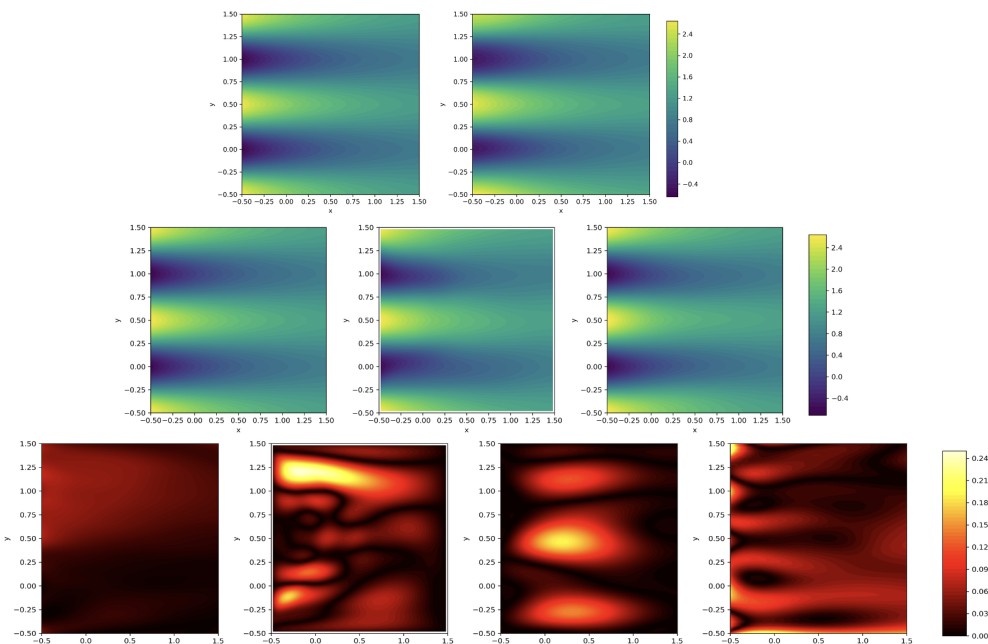

Figure 2: Results of Kovasznay flow on x-axis direction $u$. First row (from left to right): exact solution and WAN3dNS solution; Second row: Solution of $u$ from NSFnets, WAN-Biharmonic, DeepXDE; Last row: Absolute pointwise error of $u$ from WAN3DNS, NSFnets, WAN-Biharmonic and DeepXDE.

## 4.2 WAN3DNS CAN BE APPLIED TO 3D PROBLEMS (BELTRAMI FLOW)

We use the Beltrami flow Ethier & Steinman (1994); Joseph (2021) to demonstrate the performance of WAN3DNS on a smooth unsteady problem. We consider the 3D unsteady NS equations defined over the cubic domain $[-1, 1]^3$, with the temporal domain spanning $t \in [0, 1]$. The kinematic viscosity coefficient is set to $\nu = 10$, and the initial and boundary conditions are given equation 25, equation 26 and equation 27 of the Appendix. This 3D unsteady Navier-Stokes flow has the analytical solution as shown in equation 28 in the appendix.

After training, we compare the final outputs $u_1$, $u_2$, $u_3$ with the exact solutions. We further compare WAN3DNS to NSFnets and DeepXDE using the same hyperparameters (see Table 6 at Appendix). Table 9 lists the accuracy for different domain sizes and sampling quantities, showing that denser sampling generally leads to higher accuracy. However, the overall impact is relatively minor, indicating that WAN3DNS remains robust with respect to both spatial domain variation and sampling density. We provide a visual comparison for $t = 1, z = 0$, showing that WAN3DNS achieves a lower error (see Figure 3). The comparison of $u, v$ and $p$ can be seen in Appendix Fig. 6, 7 and 9. These results show that WAN3DNS can be applied to smooth unsteady problems. We exhibit the visualization of the training dynamics in Figure 8 in the appendix, showing both the primary loss and the adversarial loss. The adversarial loss exhibits fluctuations, which is theoretically expected: the adversarial network represents arbitrary test functions used to maximize the weak constraint violation. Since the maximization objective continuously adapts to challenge the current solution, the adversarial network does not necessarily converge to a fixed state, leading to non-monotonic dynamics. Importantly, these fluctuations do not indicate instability: they are a natural and desirable characteristic of minimax optimization, analogous to saddle-point dynamics in mixed variational formulations.

To ensure robust training in practice, we also applied commonly used stabilization techniques, including asymmetric learning rates, gradient clipping, and early stopping once the adversarial loss variations indicate sufficiently enforced incompressibility.

Table 2: Relative $L^2$ error and training time comparison for Beltrami flow.

|  | $u_{error}$ | $v_error$ | $w_{error}$ | $P_{error}$ | Training time (s) |
|---|---|---|---|---|---|
| DeepXDE | 0.021 | 0.018 | 0.017 | 10.54 | **4,067.37** |
| NSFnets | 0.046 | 0.073 | 0.062 | 27.66 | 9,264.38 |
| **WAN3DNS** | **0.010** | **0.008** | **0.016** | **0.011** | 6,415.76 |

Based on Beltrami flow, we perform a hyperparameter analysis of network depth, network width, learning rate, and penalty coefficient. The results are listed in Table 8 in the appendix. It shows that WAN3DNS is stable to changes in the hyperparameters listed above. For a fairer comparison, we run these experiments with different network sizes for both the WAN3DNS and baseline models. From the results in Table 11, we see that DeepXDE achieves the best velocity accuracy among all models; however, WAN3DNS significantly outperforms both DeepXDE and NSFnets in terms of pressure accuracy. In the strong-form loss, pressure enters only through its gradient, so the optimizer can reduce the momentum residual mainly by adjusting $\mathbf{u}$ while $P$ remains weakly constrained and sensitive to high-order automatic differentiation. In the weak form, pressure appears only as a variational coupling $(P, \nabla \cdot v)$ with learned test functions, avoiding pointwise instability.

We also conduct an ablation study for Beltrami flow to test the importance of each equation in the NS equations. To do this, we set 4 different cases to train the network: governing equation only; governing equation + boundary conditions; governing equation + initial condition; and all the equations. The results in Table 10 show that WAN3DNS strongly relies on the boundary conditions.

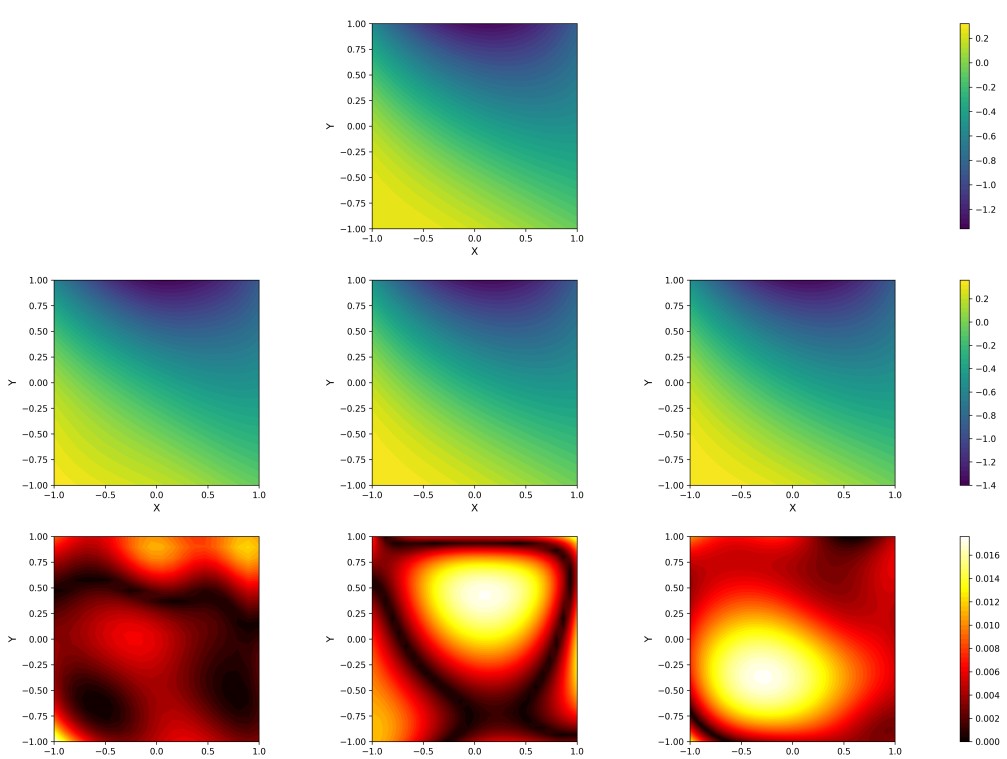

Figure 3: Results of the Beltrami flow experiment on $w$, the heatmap when $t = 1, z = 0$. First row: the exact solution of $w$; Second row: the approximate solution of WAN3DNS, NSFnets and DeepXDE; Third row: the absolute error of WAN3DNS, NSFnets and DeepXDE.

## 4.3 LID-DRIVEN CAVITY FLOW

Lid-driven cavity flow is a classical fluid dynamics problem where the flow is induced by the motion of the top wall (lid). Because this is a steady flow, $\frac{\partial u}{\partial t} = \frac{\partial v}{\partial t} = \frac{\partial w}{\partial t} = 0$ in this case. We do not need to consider the initial condition here. Slip boundary: Consider the domain $[0, 1]^3$, the velocity boundary conditions are defined as follows: on the bottom and four lateral boundaries (in total five surfaces), all velocity components (u, v, w) are set to zero. Only the top lid exhibits a unidirectional translational motion in the x-direction, specifically with a prescribed nonzero u-velocity (x-component velocity) while maintaining $v = 0$ and $w = 0$.

Although this case has a strong solution due to the low Reynold number ($Re = 400$), there exist singular points when $z = 1, x = 0$ and $z = 0, x = 1$. We take the solutions from the classical multigrid method and projection as the ground truth. Then we calculate the $L^2$ distance between each method and the ground truth. We report the $L^2$-error of these two algorithms in the Table. 3. We take $y = 0.5$ as the slice face and visualize $u$, $v$, $w$ and their absolute errors for both algorithms; see the result figure 10 in the Appendix. The PINNs baseline is implemented in TensorFlow using a standard fully connected MLP with tanh activation, following the classical PINNs formulation. Since existing frameworks do not directly support the lid-driven cavity case, we implemented this model ourselves for a fair comparison. We observe that the velocity profiles predicted by PINNs deviate substantially from classical benchmarks, particularly near boundary layers.

While PINNs have shown promise in forward modeling, their efficacy diminishes in complex flow regimes, as shown in the result figure in the Supplementary information. This discrepancy suggests that current PINN architectures struggle to enforce no-slip constraints effectively. The results of this experiment show that WAN3DNS performs better than PINNs when the solution is not smooth.

Table 3: $L^2$ error of WAN3DNS and PINNs

| $L^2$ error | $u$ | $v$ | $w$ |
|---|---|---|---|
| **WAN3DNS** | **4.990e-03** | **8.144e-08** | **4.176e-03** |
| PINNs | 2.412e-01 | 3.497e-03 | 1.648e-01 |

## 5 DISCUSSION

We propose WAN3DNS, a weak adversarial neural network algorithm for solving 2D and 3D incompressible Navier–Stokes equations within a unified framework. The method offers three main contributions: First, WAN3DNS directly enforces the weak formulation, allowing lower regularity assumptions and enhanced robustness in challenging flow settings. Second, we provide a rigorous error analysis based on Galerkin theory,thereby establishing theoretical grounding and stability guarantees during training. Third, extensive numerical evaluations on benchmark fluid dynamics problems show that WAN3DNS consistently outperforms current state-of-the-art models in both accuracy and stability.

We further state that this work is a first step toward more challenging turbulent benchmarks, which are planned for future extensions. Appendix K includes a detailed discussion of the current limitations. Due to the mathematical complexity of weak solutions in the Navier–Stokes regime, we did not directly extend our MLP-based formulation to operator-learning architectures, despite their strong generalization capability for weak solution identification. As part of our future research, we plan to integrate neural operators and graph neural networks into the proposed weak framework to improve scalability and generalization across varying flow regimes.

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

APPENDIX OUTLINE

## A  PAPER REVIEW

The first category, exemplified by PINNs Raissi et al. (2019), neural networks that are trained to solve supervised learning tasks while respecting any given laws of physics described by general nonlinear partial differential equations. The original PINNs paper Raissi et al. (2019) focused on 2D NS cases. Subsequent models, such as Hidden Fluid Mechanics (HFM) Raissi et al. (2020) and Navier-Stokes flow nets (NSFnets) Jin et al. (2021) extended this line of work to 3D NS simulations. However, they are based on the classical solution; the uniqueness of inferred velocity and pressure fields is not guaranteed without sufficient scalar gradients at the boundaries. Tools such as DeepXDE Lu et al. (2021) have incorporated specific NS problems as benchmarks for neural network-based solvers. Applications of neural network-driven NS simulations span multiple domains, such as oil reservoir modeling Meduri & Devi (2024) and medical simulation of blood flow Selby et al. (2025). $\Delta$-PINNs Costabal et al. (2024) proposes a novel positional encoding mechanism for PINNs based on the eigenfunctions of the Laplace–Beltrami operator.

The second paradigm, typified by DeepONet Lu et al. (2019), takes advantage of learning operators accurately and efficiently from a relatively small dataset. Fourier neural operator (FNO) Li et al. (2020) parameterizes the integral kernel directly and learns the resolution-invariant solution operator for the Navier-Stokes equation in the turbulent regime. Dynamic kernel Fourier Neural Operators (DSFNOs) Qiu et al. (2024) equip FNOs with operators to learn dynamic kernels. The Latent Neural Operator (LNO) Wang & Wang (2024) solves PDEs in the latent space. Geometry-Informed Operator (GINO) Li et al. (2023) uses a signed distance function (SDF) and a point cloud representation of the input shape and a neural operator based on graph and Fourier architectures to learn the solution operator. The co-domain attention neural operator (CoDA-NO) Rahman et al. (2024) tokenizes functions along the co-domain or channel space, allowing self-supervised learning or pre-training of multiple PDE systems. The derivative-enhanced deep operator network (DE-DeepONet) Qiu et al. (2024) leverages derivative information to enhance the precision of solution prediction and provides a more accurate approximation of solution-to-parameter derivatives.

## B  ALGORITHM IMPLEMENTATION

### B.1  IMPLEMENTATION

We implement WAN3DNS using Python 3.7 and Tensorflow 1.15. All experiments use Adam optimizer with mini-batch training. We provide a summary of the method as an Algorithm 1, a workflow 1, and a table with the Hyperparameters used in this work.

**Algorithm 1:** WAN3DNS

**Input:** $N_r/N_b/N_I; \Omega \times [0, T]$.
**Output:** Weak solution $u_\theta(x, y, z, t)$.
1 **Initialize:** Network architecture $u_\theta : \Omega \times [0, T] \to \mathbb{R}^3, S_\eta, v_\eta : \Omega \to \mathbb{R}^3$ and parameters $\theta, \eta$;
   Number of steps $K$; learning rate $\tau_\theta, \tau_\eta$ .
2 **for** i=1,...,K **do**
3      **for** $k = 1, ..., k_u$ **do**
4         $\theta \leftarrow \theta - \tau_\theta \nabla_\theta L_u$;
5      **for** $k = 1, ..., k_T$ **do**
6         $\eta \leftarrow \eta - \tau_\eta \nabla_\eta L_T$.

### B.2 HYPERPARAMETERS

Table 4: Hyperparameters for the three datasets. $\lambda$ means the learning rate and $\alpha$ means the penalty coefficient for initial and boundary conditions.

| Dataset | Architecture | #Epochs | $\lambda$ | $\alpha$ |
|---|---|---|---|---|
| Kovasznay Flow | [2, 50, 50, 50, 50, 50, 50, 3] | 20, 000 | 0.001 | 10, 000 |
| Beltrami Flow | [4, 50, 50, 50, 50, 50, 50, 4] | 20, 000 | 0.001 | 100 |
| Lid-driven cavity Flow | [3, 40, 40, 40, 40, 40, 40, 40, 40, 4] | 20, 000 | 0.001 | 10, 000 |

Table 5: Hyperparameters for the experiments in implementation.

| Symbol | Hyperparameter |
|---|---|
| $N_r$ | Number of samplings inside the domain |
| $N_b$ | Number of samplings on the boundary |
| $N_I$ | Number of samplings at initial condition |
| $K$ | Number of epoches |
| $k_\theta$ | The number of iterations of primary network in each epoch |
| $k_\eta$ | The number of iterations of adversary network in each epoch |
| layer $\times$ nodes | Primary network or Adversary network |
| $\alpha$ | Penalty coefficient for boundary condition |
| $\beta$ | Penalty coefficient for initial condition |
| $\nu$ | Viscosity coefficient (1/Reynold number) |

Table 6: Dataset Properties

| Dataset | Problem Type | #Points | # Initial | # Boundaries | #Test Points |
|---|---|---|---|---|---|
| Kovasznay | 2D steady | 2600 | - | 400 | 10,000 |
| Beltrami | 3D evolutionary | 20,000 | 5,000 | 4,000 | 10,000 |
| Lid-driven cavity | 3D steady | 10,000 | - | 6,000 | 10,000 |

## C    PROOF OF THE THEOREM

To prove Theorem 1, we provide the definition of bilinear and trilinear forms

$$a(\mathbf{u}, \mathbf{v}) = \sum_{i,j=1}^{n} \int \frac{\partial \mathbf{u}_j}{\partial x_i} \frac{\partial \mathbf{v}_j}{\partial x_i} dx, \quad b(\mathbf{u}, \mathbf{v}, \mathbf{w}) = \sum_{i,k=1}^{n} \int_{\Omega} \mathbf{u}_k (D_k \mathbf{v}_i) \mathbf{w}_i dx,$$

and Gronwall inequality:

**Theorem 2 (Gronwall inequality)** *Let $\xi(t)$ be a nonnegative integrable function on $[0,T]$, and suppose that for a.e. $t \in [0,T]$,*

$$\xi(t) \leq C_1 \int_0^t \xi(s)\, ds + C_2,$$

*where $C_1, C_2 > 0$ are constants. Then we have*

$$\xi(t) \leq C_2 \left(1 + C_1 t e^{C_1 t}\right), \quad \text{for a.e. } t \in [0,T].$$

Let $\mathbf{u}$ and $\mathbf{u}_\theta$ be the exact solution and approximate velocity, and define the error as $\hat{\mathbf{u}} = \mathbf{u} - \mathbf{u}_\theta$. $S$ and $\mathbf{v}$ are test functions as defined before. Now we prove Theorem 1. The network residuals can be expressed as:

$$\begin{aligned}
R_{PDE} &= [(\frac{\partial \mathbf{u}}{\partial t}, v) - \nu(\nabla \mathbf{u}, \nabla \mathbf{v}) + b(\mathbf{u}, \mathbf{u}, \mathbf{v}) - (P, \nabla \cdot \mathbf{v})] \\
&\quad - [(\frac{\partial \mathbf{u}_\theta}{\partial t}, \mathbf{v}) - \nu(\nabla \mathbf{u}_\theta, \nabla \mathbf{v}) + b(\mathbf{u}_\theta, \mathbf{u}_\theta, \mathbf{v}) - (P, \nabla \cdot \mathbf{v})] \\
&= (\hat{\mathbf{u}}', \mathbf{v}) + \nu(\nabla \hat{\mathbf{u}}, \nabla \mathbf{v}) + b(\hat{\mathbf{u}}, \mathbf{u}, \mathbf{v}) + b(\mathbf{u}, \hat{\mathbf{u}}, \mathbf{v}) - b(\hat{\mathbf{u}}, \hat{\mathbf{u}}, \mathbf{v}) - (\hat{P}, \nabla \cdot \mathbf{v}),
\end{aligned} \tag{17}$$

$$R_{div} = (\hat{\mathbf{u}}, S), \quad R_t(x) = \hat{\mathbf{u}}(t=0) \quad R_s = \mathbf{u}_\theta(x,t) - \mathbf{u}_\Gamma(x,t)$$
$$).$$

where $\hat{\mathbf{u}}'$ represents the derivative of $\hat{\mathbf{u}}$ with respect to $t$. Since $\|\mathbf{v}\| \leq 1$ and $\|\frac{\hat{\mathbf{u}}}{\|\hat{\mathbf{u}}\|}\| = 1$, we have:

$$\begin{aligned}
R_{PDE} &\leqslant (\hat{\mathbf{u}}', \frac{\hat{\mathbf{u}}(t)}{\|\hat{\mathbf{u}}\|}) + \nu a(\hat{u}, \frac{\hat{u}}{\|\hat{u}(t)\|}) + b(\hat{u}, u, \frac{\hat{u}(t)}{\|\hat{u}(t)\|}) \\
&\quad + b(u, \hat{u}, \frac{\hat{u}(t)}{\|\hat{u}(t)\|}) - b(\hat{u}, \hat{u}, \frac{\hat{u}(t)}{\|\hat{u}(t)\|}) \\
&= -\frac{1}{2} \partial_t \|\hat{u}\|_2 + \frac{\nu}{\|\hat{u}\|} a(\hat{u}, \hat{u}) + \frac{1}{\|\hat{u}\|} b(\hat{u}, u, \hat{u}) + (\frac{\hat{\mathbf{u}}}{\|\mathbf{u}\|} \cdot \nabla)\hat{P}
\end{aligned} \tag{18}$$

where the equal sign is because $b(u, \hat{u}, \hat{u}) = b(\hat{u}, \hat{u}, \hat{u}) = 0$. Integrating over the domain $D$ and applying integration by parts, we get the following.

$$\begin{aligned}
2\int_D R_{PDE} dx &\leqslant -\frac{d}{dt} \int_D \|\hat{u}\|_2 dx + \frac{2\nu}{\|\hat{u}\|} \int \|\nabla \hat{u}_j\|_2^2 dx - \frac{2\nu}{\|\hat{u}\|} \int_{\partial D} \hat{u}_j (\hat{n} \cdot \nabla \hat{u}_j) ds(x) \\
&\quad - \frac{2}{\|\hat{u}\|} \int_D \hat{u}((\hat{u} \cdot \nabla)u) dx + \int_D R_{div}(\|\hat{u}\|_2^2 + 2\hat{P}) dx - \int_{\partial D} \hat{u} \cdot \hat{n} \cdot \|u\|_2 ds(x)
\end{aligned} \tag{19}$$

which can also be written as

$$\begin{aligned}
\frac{d}{dt} \int \|\hat{\mathbf{u}}\|_2^2 dx &\leqslant 2\nu \sum_{j=1}^{d} \int \|\nabla \hat{u}_j\|_2^2 dx - 2\nu \int_{\partial D} \hat{u}_j (\hat{n} \cdot \nabla \hat{u}_j) ds(x) - 2 \int_D \hat{u}((\hat{u} \cdot \nabla)u) dx \\
&\quad + \int_D R_{div}(\|\hat{u}\|_2^2 + 2\hat{P}) dx - \int_{\partial D} \hat{u} \cdot \hat{n} \cdot \|u\|_2^2 ds(x) - 2 \int_D R_{PDE} \|\hat{u}(t)\| dx
\end{aligned} \tag{20}$$

Furthermore, for a constant $C_1(\|u\|_{C^1}, \|\hat{u}\|_{C^1})$, the following inequalities hold:

$$
\begin{aligned}
|\int_{\partial D} \hat{u}_j (\mathbf{n} \cdot \nabla \hat{u}_j) ds(x)| &\leqslant C_1(\|R_s\|_{L^1(\partial D)}), \\
-\int_D \hat{u}((\hat{u} \cdot \nabla)u) dx &\leqslant d^2 \|\nabla u\|_{L^\infty(D)} \int_D \|\hat{u}\|_2^2 dx, \\
|\int_{\partial D} \hat{\mathbf{u}} \cdot \hat{\mathbf{n}} \cdot \|\mathbf{u}\|_2^2 ds(x)| &\leqslant C_1 \|R_s\|_{L^1(\partial D)}.
\end{aligned}
\tag{21}
$$

Integrating over the time interval $[0, \tau] \subset [0, T]$, we obtain:

$$
\begin{aligned}
\int_f \|\hat{u}(x,\tau)\|_2^2 dx \leqslant & \|R_t\|_{L^2(D)}^2 + C_1 \sqrt{T|D|} \|R_{div}\|_{L^2(D)} + \|R_{PDE}\|_{L^2}^2 \\
& + (2d^2 \|\nabla u\|_{L^\infty(D)} + 1) \int_{D \times [0,T]} \|\hat{u}(x,t)\|_2^2 dx dt \\
& + C_1(1+\nu)\sqrt{T|\partial D|}\|R_s\|_{L^2(\partial D)}
\end{aligned}
\tag{22}
$$

Using Gronwall's inequality 2, we have:

$$
\int_D \|\hat{u}(x,t)\|_2^2 dx dt \leqslant C(2d^2 \|\nabla u\|_{L^\infty(D)} + 1)Te^{(2d^2\|\nabla u\|_{L^\infty(D)}+1)T},
\tag{23}
$$

where:

$$
\begin{aligned}
C = & \|R_t\|_{L^2(D)}^2 + C_1\sqrt{T|D|}\|R_{div}\|_{L^2(D)} \\
& + C_1(1+\nu)\sqrt{T|\partial D|}\|R_s\|_{L^2(\partial D \times [0,T])} + \|R_{PDE}\|_{L^2}^2.
\end{aligned}
\tag{24}
$$

# D  BASELINES

**What is the strength/weakness?**  DeepXDE demonstrates notable robustness and versatility, supporting forward problems, inverse problems, operator learning, and multi-fidelity learning. Its strength lies in the implementation of residual-based adaptive refinement (RAR), which enhances training efficiency, and its capability to handle multiphysics problems and complex geometry domains through constructive solid geometry (CSG), thereby reducing the need for extensive computational geometry preprocessing. However, a key weakness is the empirical selection of neural network architectures, which currently relies heavily on user experience rather than systematic optimization. Additionally, unlike traditional numerical methods, PINNs lack a priori error bounds, which limits their reliability in certain applications.

NSFNets offers a comparative analysis of two mathematical formulations—Velocity-Pressure (VP) and Vorticity-Velocity (VV)—within the PINN framework. A significant strength is its ability to sustain turbulent flow in subdomains using only DNS data at the boundaries, addressing a major challenge for data-free PINNs in modeling complex flows. Nevertheless, this approach incurs higher computational costs and exhibits less robustness compared to traditional numerical methods for standard cases.

WAN-Biharmonic introduces a weak adversarial network (WAN) approach for the biharmonic formulation of the 2D Navier-Stokes equations, achieving high accuracy even for flows lacking strong solutions. Its strength lies in effectively handling such challenging scenarios using a stream function derived from the velocity field. However, a primary limitation is its restriction to 2D problems, as the method cannot be directly extended to 3D due to the absence of a scalar function that fully describes the velocity field while satisfying the incompressibility condition.

**What were they tested on before?**  DeepXDE has been validated on various benchmarks, including the Kovasznay flow, as documented on its official website (https://deepxde.readthedocs.io/en/latest/). Similarly, NSFnets have been tested on several canonical flows, such as Kovasznay and Beltrami flows. WAN-Biharmonic has been evaluated in Kovasznay flow and 2D lid-driven cavity flow, demonstrating its efficacy in these settings.

**Why use these 3 baselines for the first example, 2 baseline models for the second example, and just 1 baseline model for the last one?** Given that WAN-Biharmonic is specifically designed for 2D problems, it is used solely as a benchmark for the Kovasznay flow in the first example. For the Beltrami flow analysis in the second example, which may involve 3D characteristics, DeepXDE and NSFnets are selected as baseline models due to their support for both 2D and 3D simulations and their established robustness in handling a wider range of flow conditions. For 3D Lid-driven cavity flow, traditional PINNs fail at singular points due to strong form derivation, so there are not many models for this flow. In contrast, WAN3DNS's weak form naturally avoids this problem.

## E   MORE INFORMATION ABOUT KOVASZNAY FLOW

Fig. 4 and 5 visualize the comparison with other algorithms when the neural size is $6 \times 50$. Table 7 shows the comparison results when the neural size is $9 \times 30$.

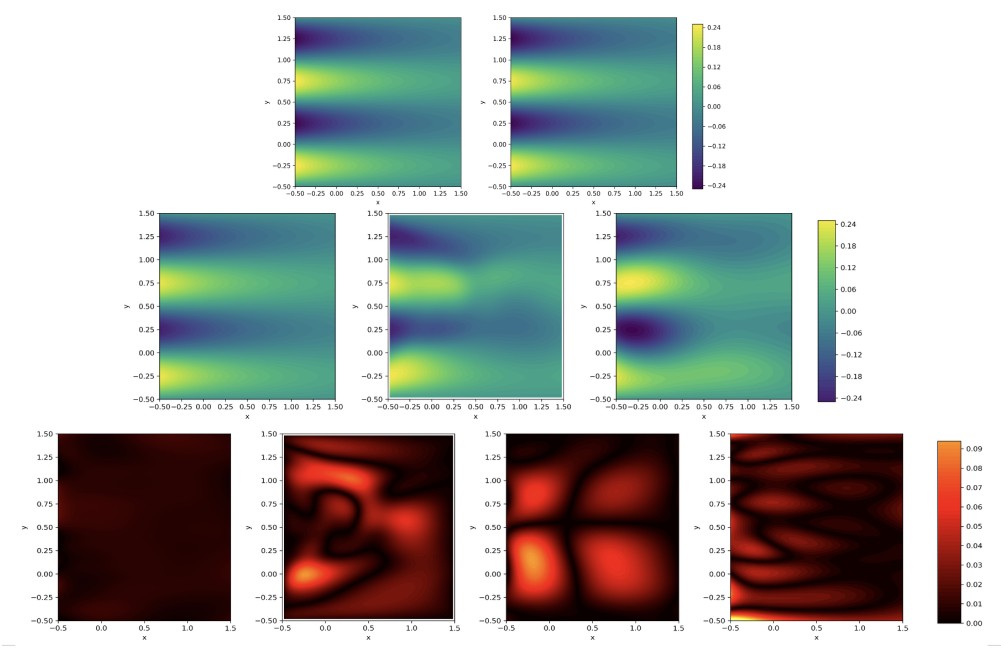

Figure 4: Results of Kovasznay flow on y-axis direction $v$. First row (from left to right): exact solution and WAN3dNS solution; Second row: Solution of $v$ from NSFnets, WAN-Biharmonic, DeepXDE; Last row: Absolute pointwise error of $v$ from WAN3DNS, NSFnets, WAN-Biharmonic and DeepXDE.

Table 7: Results of Kovasznay flow when the neural network has 9 layers, and 30 neurons per layer. Performance remains below the $6 \times 50$ neural network baseline but is consistently higher than that of other algorithms within the same architecture.

|  | $u_{error}$ | $v_{error}$ | $\psi_{error}$ | $P_{error}$ | Training time (s) |
|---|---|---|---|---|---|
| WAN-Biharmonic | 0.040 | 0.223 | 0.015 | - | 815.814 |
| DeepXDE | 0.009 | 0.043 | - | **0.012** | 910.566 |
| NSFnets | 0.077 | 0.207 | - | 4.070 | 899.771 |
| **WAN3DNS** | **0.005** | **0.025** | - | 0.039 | **728.065** |

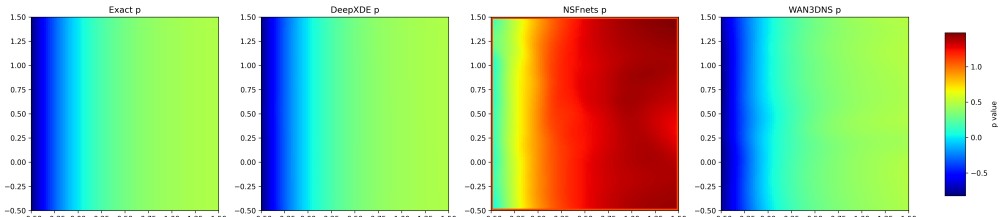

Figure 5: Results of Kovasznay flow on pressure $p$. DeepXDE performs better than WAN3DNS, but the NSFnets totally loss it's effectiveness.

## F   More information about Beltrami flow

The initial boundary conditions of $u$, $v$ and $w$ are Eq.equation 25, Eq. equation 26 and Eq.equation 27, respectively:

$$\begin{cases} u(0,y,z,t) = -[\sin(y+z) + e^z \cos y]e^{-t}, \\ u(1,y,z,t) = -[e \sin(y+z) + e^z \cos(1+y)]e^{-t}, \\ u(x,0,z,t) = -[e^x \sin z + e^z \cos x]e^{-t}, \\ u(x,1,z,t) = -[e^x \sin(1+z) + e^z \cos(1+x)]e^{-t}, \\ u(x,y,0,t) = -[e^x \sin y + \cos(x+y)]e^{-t}, \\ u(x,y,1,t) = -[e^x \sin(1+y) + e \cos(y+x)]e^{-t}, \\ u(x,y,z,0) = -e^x \sin(y+z) - e^z \cos(x+y). \end{cases} \quad (25)$$

$$\begin{cases} v(0,y,z,t) = -[e^y \sin z + \cos(y+z)]e^{-t}, \\ u_2(1,y,z,t) = -[e^y \sin(1+z) + e \cos(z+y)]e^{-t}, \\ u_2(x,0,z,t) = -[\sin(z+x) + e^x \cos z]e^{-t}, \\ u_2(x,1,z,t) = -[e \sin(x+z) + e^x \cos(1+z)]e^{-t}, \\ u_2(x,y,0,t) = -[e^y \sin x + e^x \cos y]e^{-t}, \\ u_2(x,y,1,t) = -[e^y \sin(1+x) + e^x \cos(y+1)]e^{-t}, \\ v(x,y,z,0) = -e^y \sin(z+x) - e^x \cos(y+z). \end{cases} \quad (26)$$

$$\begin{cases} w(0,y,z,t) = -[e^z \sin y + e^y \cos z]e^{-t}, \\ w(1,y,z,t) = -[e^z \sin(1+y) + e^y \cos(z+1)]e^{-t}, \\ w(x,0,z,t) = -[e^z \sin x + \cos(z+x)]e^{-t}, \\ w(x,1,z,t) = -[e^z \sin(x+1) + e \cos(x+z)]e^{-t}, \\ w(x,y,0,t) = -[\sin(x+y) + e^y \cos x]e^{-t}, \\ w(x,y,1,t) = -[e \sin(y+x) + e^y \cos(x+1)]e^{-t}, \\ w(x,y,z,0) = e^z \sin(x+y) + e^y \cos(z+x). \end{cases} \quad (27)$$

$$\begin{aligned} u(x,y,z,t) &= -[e^x sin(y+z) + e^z cos(x+y)]e^{-t}, \\ v(x,y,z,t) &= -[e^y sin(z+x) + e^x cos(y+z)]e^{-t}, \\ w(x,y,z,t) &= -[e^z sin(x+y) + e^y cos(z+x)]e^{-t}, \\ P(x,y,z,t) &= -\frac{1}{2}[e^{2x} + e^{2y} + e^{2z} + 2sin(x+y)cos(z+x)e^{y+z} \\ &\quad + 2sin(y+z)cos(x+y)e^{y+z} + 2sin(z+x)cos(y+z)e^{x+y}]e^{-2t}. \end{aligned} \quad (28)$$

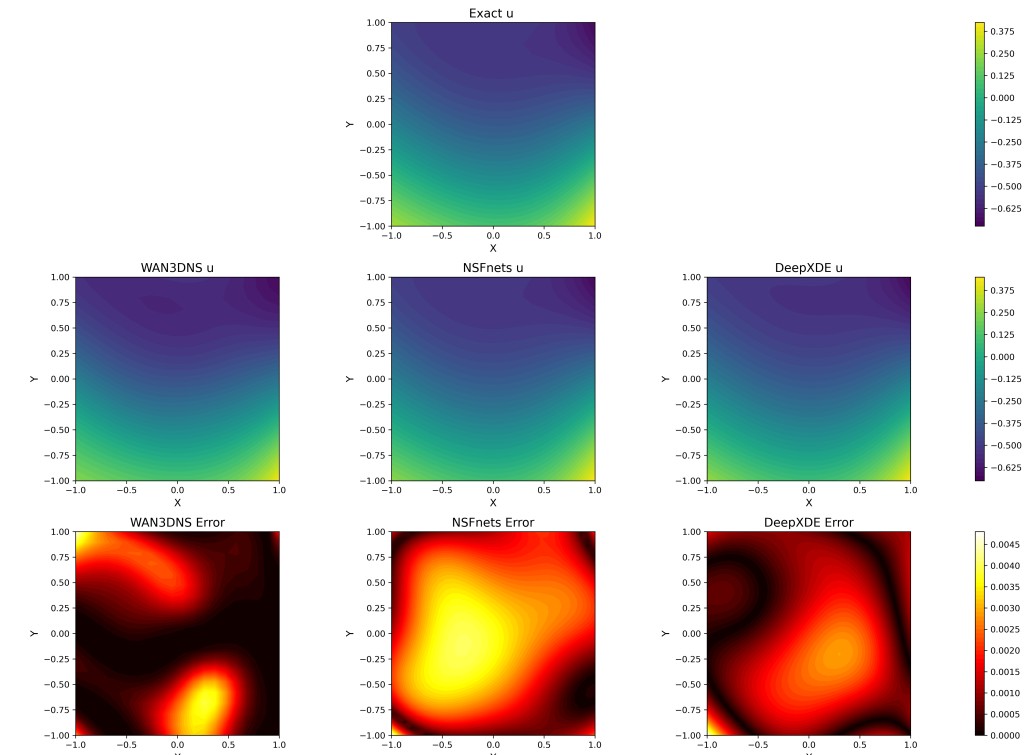

Figure 6: Results of the Beltrami flow experiment on $u$ when $t = 1, z = 0$. First row: the exact solution of $u$; Second row (from left to right): the approximate solution of WAN3DNS, NSFnets and DeepXDE; Third row (from left to right): the absolute error from WAN3DNS, NSFnets and DeepXDE. Although WAN3DNS has a smaller overall error, DeepXDE's results are more stable, and NSFnets has the worst performance.

## G   HYPERPARAMETER SENSITIVITY ANALYSIS

To test the sensitivity of Algorithm 1, we sweep different hyperparameters (number of layers, number of neurons per layer, penalty coefficient of initial and boundary conditions, learning rate) for a sensitivity analysis. The results are presented in Table 8. From the table, we can observe that as the learning rate increases, the error decreases. However, the error does not show a clear correlation with the number of network layers, the number of neurons, or the penalty coefficient. The best performance is achieved with the parameters: 6 layers × 50 neurons, learning rate = 0.001 and penalty coefficient = 100.

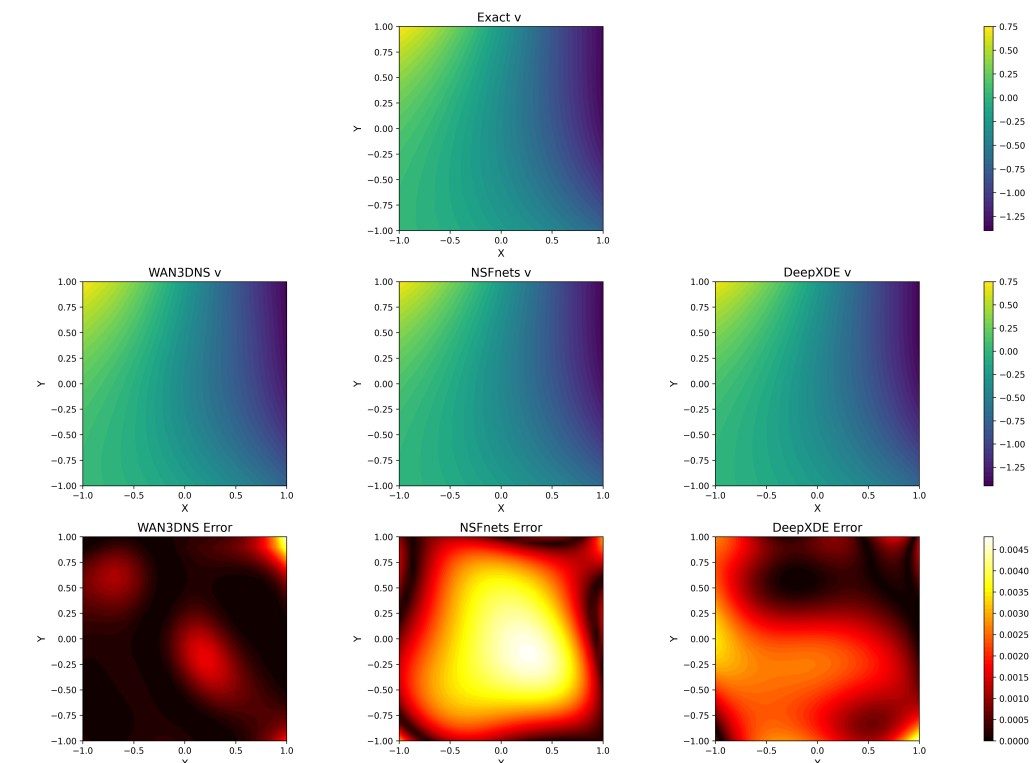

Figure 7: Results of the Beltrami flow experiment on $v$ when $t = 1, z = 0$. First row: the exact solution of $v$; Second row (from left to right): the approximate solution of WAN3DNS, NSFnets and DeepXDE; Third row (from left to right): the absolute error from WAN3DNS, NSFnets and DeepXDE. WAN3DNS shows small absolute error in most areas, DeepXDE second and NSFnets last.

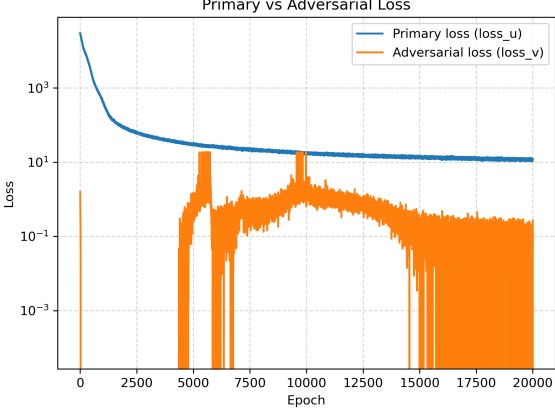

Figure 8: Loss curve of the primary network and the adversarial network. The primary loss decreases steadily, showing stable convergence of the solution, while the adversarial loss fluctuates as expected in GAN-style training before gradually reducing, indicating a balanced and stable adversarial optimization.

To assess how WAN3DNS scales with respect to the **spatial domain size** and the **sampling density**, we conducted a systematic study on the 3D Beltrami flow benchmark. Specifically, we varied:

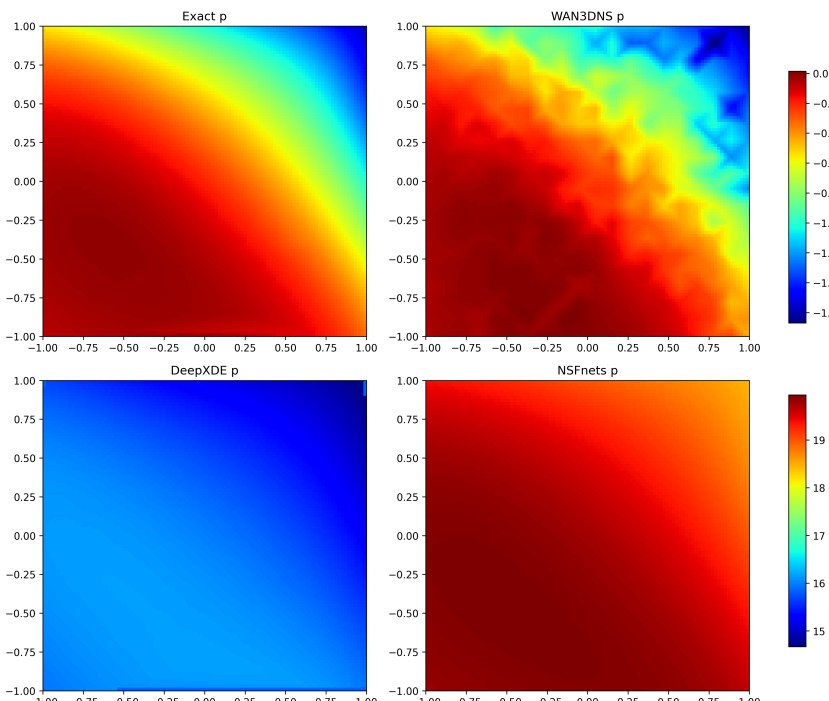

Figure 9: Results of the Beltrami flow experiment on $p$. First row (from left to right): exact solution and WAN3DNS solution; Second row (from left to right): solution from NSFnets and DeepXDE. The $p$ of WAN3DNS simulated is close to the true solution, while the results of DeepXDE and NSFnets are distorted.

- **Domain size:** (spatial cube, time fixed to ($[0, 1]$)): $\Omega_1 = [-0.5, 0.5]^3$, $\Omega_2 = [-1, 1]^3$, $\Omega_3 = [-2, 2]^3$
- **Number of interior training samples:** $N_{dm} = 10{,}000;\ 20{,}000;\ 30{,}000$.

We observe that across all spatial domains, increasing the number of interior training samples consistently reduces the velocity and pressure errors, confirming that denser collocation improves the weak enforcement of the governing equations. Conversely, enlarging the spatial domain while keeping the number of samples fixed results in noticeable error growth, particularly in pressure accuracy for the $[-2, 2]^3$ case, due to sparsity in the sampling distribution and the associated degradation of solution resolution. Overall, the findings demonstrate that WAN3DNS is sensitive to the sampling density relative to the spatial extent of the flow domain—**the sparser the sampling points are, the larger the error and the lower the accuracy**—whereas the absolute domain size alone does not significantly affect performance as long as sufficient training samples are provided to maintain spatial resolution.

## H  ABLATION STUDY

The configuration of the loss function is pivotal in physics-informed learning. To quantitatively probe the impact of initial and boundary conditions on the solution fidelity of the Navier-Stokes (NS) equations, we evaluated four specific formulations of the loss function:

- Case 1 - PDE Residuals only (PDE-only): The loss is computed solely based on the residuals of the governing equations within the domain.
- Case 2 - PDE and Boundary Conditions (PDE+BC): The loss includes the PDE residuals plus the errors on the spatial boundaries.
- Case 3 - PDE and Initial Conditions (PDE+IC): The loss includes the PDE residuals plus the errors at the initial time.

Table 8: Hyperparameter sensitivity of WAN3DNS for the Beltrami flow: Penalty coefficient is more influential than network architecture. The optimal performance is achieved with a penalty coefficient of 100 and a 6×50 network structure, highlighting the importance of constraint enforcement. **Bold** values indicate the best performance in each parameter group, while *italic* values highlight the parameter being varied within each test block.

| Neural Architecture | Learning Rate | Penalty Coefficient | $u_{error}$ | $v_{error}$ | $w_{error}$ | $P_{error}$ |
|---|---|---|---|---|---|---|
| **Network Depth (Layers)** | | | | | | |
| *4 × 50* | 0.001 | 100 | 0.026 | 0.025 | 0.029 | 0.021 |
| *6 × 50* | 0.001 | 100 | **0.010** | **0.008** | **0.016** | **0.011** |
| *8 × 50* | 0.001 | 100 | 0.033 | 0.034 | 0.027 | 0.017 |
| **Layer Width (Neurons per Layer)** | | | | | | |
| *6 × 30* | 0.001 | 100 | 0.036 | 0.035 | 0.032 | 0.028 |
| *6 × 50* | 0.001 | 100 | **0.010** | **0.008** | **0.016** | **0.011** |
| *6 × 70* | 0.001 | 100 | 0.030 | 0.021 | 0.021 | 0.024 |
| **Learning Rate** | | | | | | |
| 6 × 50 | *1e-4* | 100 | 0.023 | 0.023 | 0.023 | 0.013 |
| 6 × 50 | *5e-4* | 100 | 0.022 | 0.020 | 0.020 | 0.018 |
| 6 × 50 | *0.001* | 100 | **0.010** | **0.008** | **0.016** | **0.011** |
| **Penalty Coefficient** | | | | | | |
| 6 × 50 | 0.001 | *10* | 0.022 | 0.019 | 0.020 | 0.012 |
| 6 × 50 | 0.001 | *100* | **0.010** | **0.008** | **0.016** | **0.011** |
| 6 × 50 | 0.001 | *1000* | 0.033 | 0.032 | 0.031 | 0.035 |

Table 9: Impact of domain size and sampling density on WAN3DNS performance. **Take home message:** The sparser the sampling points during training, the larger the error and the lower the accuracy.

| Domain $\Omega$ | $N_{dm}$ | $\|u - u^*\|_{L^2}$ | $\|v - v^*\|_{L^2}$ | $\|w - w^*\|_{L^2}$ | $\|p - p^*\|_{L^2}$ |
|---|---|---|---|---|---|
| $[-0.5, 0.5]^3$ | 10,000 | 0.016 | 0.019 | 0.018 | 0.013 |
| $[-0.5, 0.5]^3$ | 20,000 | 0.014 | 0.014 | 0.019 | 0.013 |
| $[-0.5, 0.5]^3$ | 30,000 | 0.001 | 0.002 | 0.003 | 0.011 |
| $[-1, 1]^3$ | 10,000 | 0.027 | 0.029 | 0.028 | 0.013 |
| $[-1, 1]^3$ | 20,000 | 0.010 | 0.008 | 0.016 | 0.011 |
| $[-1, 1]^3$ | 30,000 | 0.006 | 0.004 | 0.003 | 0.009 |
| $[-2, 2]^3$ | 10,000 | 0.191 | 0.213 | 0.192 | 0.129 |
| $[-2, 2]^3$ | 20,000 | 0.190 | 0.208 | 0.164 | 0.103 |
| $[-2, 2]^3$ | 30,000 | 0.108 | 0.109 | 0.102 | 0.034 |

- Case 4 - PDE with Full Constraints (PDE+IC+BC): The loss incorporates the residuals of the PDEs, the initial conditions, and the boundary conditions, representing the well-posed problem.

The numerical errors associated with each of these configurations are presented in Table 10.

The results lead to a critical conclusion: Boundary conditions are indispensable for the NS equations. As evidenced by the significantly larger errors in Case 1 and Case 3 (see Table 10), the absence of boundary constraints renders the fluid dynamics problem ill-posed. A solution driven only by the PDEs and initial conditions (Case 3) is fundamentally unstable; the velocity and pressure fields may develop unphysical instabilities and diverge arbitrarily over time. This is because boundary conditions act as spatial constraints that "anchor" the solution, ensuring that it respects the physical confines of the domain. Without them, even an accurate initial state is insufficient to guarantee a meaningful

Table 10: Ablation study: Relative $L^2$ error for different loss functions.

| Loss Function | $u_{error}$ | $v_{error}$ | $w_{error}$ | $P_{error}$ |
|---|---|---|---|---|
| PDE | 0.964 | 0.869 | 1.026 | 0.021 |
| PDE+BC | 0.029 | 0.028 | 0.029 | 0.031 |
| PDE+IC | 0.677 | 0.891 | 0.916 | 1.024 |
| PDE+BC+IC (Full setup) | **0.010** | **0.008** | **0.016** | **0.011** |

solution, as the fluid motion is not uniquely determined and can evolve towards non-physical states. Therefore, the inclusion of correct boundary conditions is not merely beneficial but essential for the model to capture the true physics of fluid flow.

## I    MORE INFORMATION ABOUT LID-DRIVEN CAVITY FLOW

The visualization of the results of the lid-driven cavity flow can be seen in Fig. 10.

## J    IMPLEMENTATION

The adversarial network is initialized or regularized as following:

- All weights of network are initialized with Xavier-uniform initialization, and all biases are initialized to zero.
- To enforce the correct functional structure, we multiply the raw network output by a fixed compactly supported bump function.

For numerical stability during the training, we use several safeguards:

- Safe gradient computation.
- Robust loss design for the adversarial network.
- Gradient clipping and handling Non-gradients.
- NaN/Inf checks in the loss.
- Moderate learning rates and optimizer choice.
- Early stopping.

## K    LIMITATIONS

Although the adversarial training framework of WAN3DNS introduces well-known challenges, such as increased computational costs and sensitivity to hyperparameters, this process involved sweeping key factors, including network architectures, learning rates, and penalty weights. The configurations reported in Section 4 represent the robust choices identified through this analysis, thus supporting our claims and providing practical guidance for replication.

The paper just studied the fully connected neural network as the primary and adversary network. In the future, we can refer to XNODE-WAN Oliva et al. (2022), referring to PDE as the ODE of time $t$, construct the primary network as Neural Ordinary Differential Equation (NODE) **?**.

The algorithm 1 is sensitive to hyperparameters according to Table 11. In the Beltrami flow, if we set different $\nu$, or set different hyperparameters, the performance of DeepXDE may be better than ours. We tested the performance of two other models with a neural network structure of 9×30 layers, and the comparative results are displayed in the Table 11. These results indicate that, in certain cases, DeepXDE yields partially better results than our model; however, when considered from a global perspective, our model demonstrates superior overall performance.

In Table 11, we present a comparison using a fixed 9-layer, 30-neuron architecture that is not the optimal configuration for any method. This example serves to illustrate that even with suboptimal

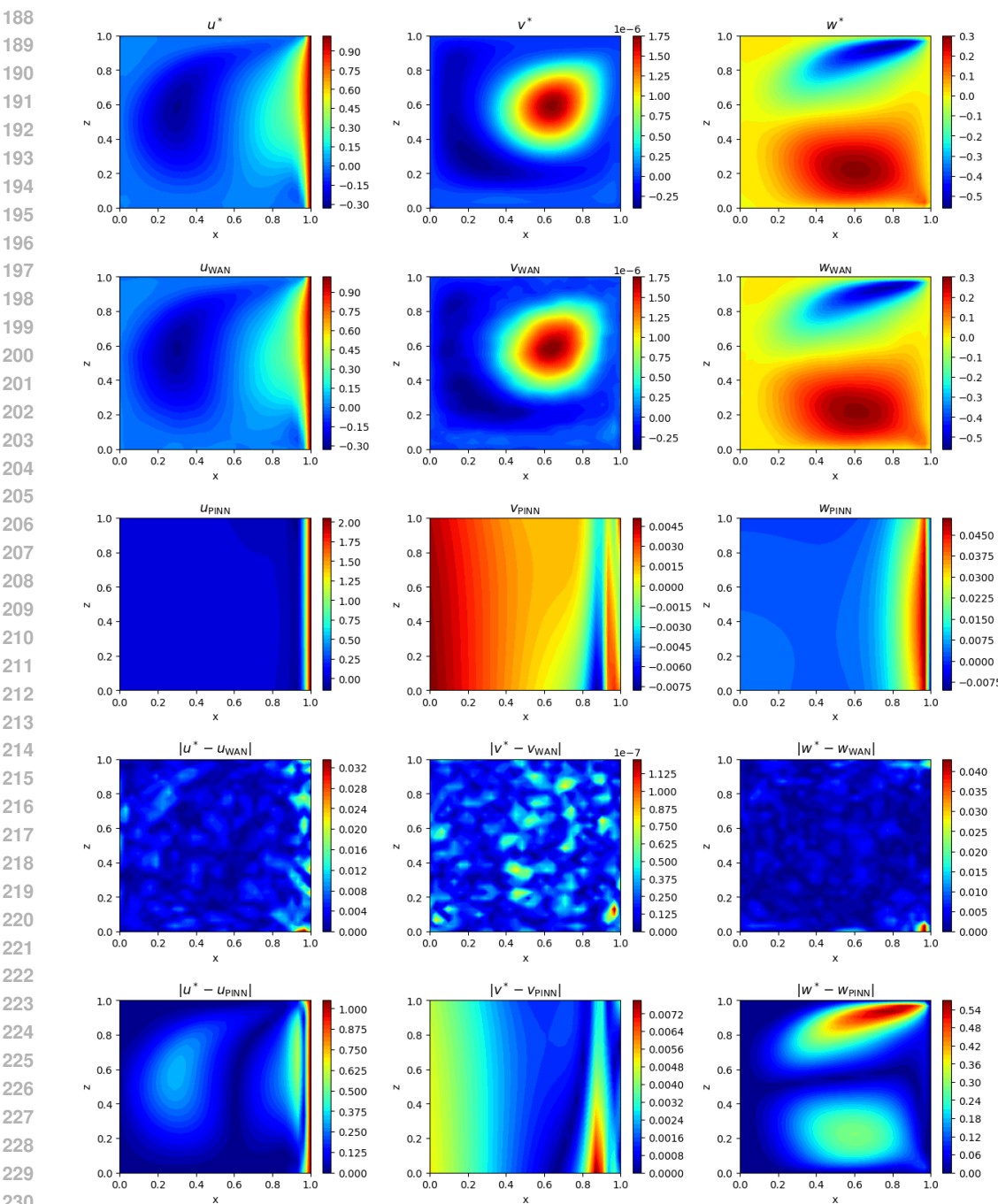

Figure 10: Results of Lid-driven cavity. Line 1 is the curve of exact solution on $y = 0.5$, line 2 the solution from WAN3DNS, line 3: the solution from PINNs, line 4: absolute error of WAN3DNS, line 5: absolute error of PINNs. The results of WAN3DNS are close to the true solution, while the results of PINNs are distorted.

hyperparameters, WAN3DNS demonstrates superior stability across all output fields, particularly in pressure accuracy. Although DeepXDE achieves lower velocity errors at these specific settings, its pressure error is orders of magnitude higher, highlighting WAN3DNS's balanced performance profile.

Table 11: Performance comparison on Beltrami flow using a fixed 9-layer network (30 neurons per layer) - not the optimal hyperparameters for any method. This example demonstrates that even with non-optimal architecture, WAN3DNS maintains balanced performance across all variables, while DeepXDE shows significantly worse pressure accuracy. At these specific hyperparameters, DeepXDE achieves better velocity errors but fails to maintain pressure accuracy.

|  | $u_{error}$ | $v_{error}$ | $w_{error}$ | $P_{error}$ | Training time (s) |
|---|---|---|---|---|---|
| DeepXDE | **0.009** | **0.017** | **0.013** | 29.94 | 35,771.238 |
| NSFnets | 0.049 | 0.030 | 0.033 | 3.27 | 12,674.357 |
| **WAN3DNS** | 0.033 | 0.034 | 0.028 | **0.018** | **7,587.387** |

## L  LLM USAGE STATEMENT

Large Language Models (LLMs) were utilized in this work to assist with grammar refinement, clarity enhancement, and code optimization. In particular, LLMs supported the editing of the manuscript by improving linguistic precision and readability. For programming-related tasks, the authors developed the core logic and structural design, and LLMs were employed to enhance code efficiency, debug routines, and refine implementation consistency. All conceptual contributions, methodological developments, and experimental setups were conceived and executed solely by the authors.

## M  REPRODUCIBILITY STATEMENT

All the source code necessary to reproduce our results are available in the link: `https://github.com/Wenran-Li/WAN3DNS`.

## N  ETHICS STATEMENT

This work focuses on developing machine learning methods for solving partial differential equations, specifically the 3D incompressible Navier-Stokes equations. The primary goal is to improve the accuracy and scalability of data-driven physical simulations, which can be beneficial in fields such as fluid dynamics, climate modeling, and engineering.

No human subjects, personally identifiable information, or sensitive data were used in this study. The methods proposed do not raise foreseeable ethical concerns related to fairness, safety, or misuse. However, as with any general-purpose learning algorithm, care should be taken when applying the proposed method to real-world systems with high safety or regulatory requirements.

The authors encourage responsible use of this research in accordance with ethical standards for scientific computing and machine learning, especially when deploying such models in sensitive or safety-critical domains.

