# OpenReview forum: "WAN3DNS: WEAK ADVERSARIAL NETWORKS FOR SOLVING 3D INCOMPRESSIBLE NAVIER-STOKES EQUATIONS"
_ICLR.cc/2026/Conference — Submitted to ICLR 2026_

### Official Review · Reviewer_fsru · 2025-10-17

**Soundness:** 3
**Presentation:** 2
**Contribution:** 2
**Rating:** 6
**Confidence:** 2

**Summary:**

This manuscript introduces the WAN3DNS framework, which is a weak-form adversarial NN framework to solve 2D/3D incompressible N.-S. equations. In contrast to classical PINNs that rely on differentiable strong solutions, the proposed WAN3DNS framework operates directly on the weak formulation of the governing equations. It reformulates the PDE system as a min-max optimization problem, where the primal network predicts velocity and pressure, and the adversarial network generates divergence-free test functions to enforce the PDE constraints. Incompressibility is implicitly enforced under this new formulation. A theoretical error bound links the L2 training error to physical accuracy. Empirical validation on three canonical benchmarks shows that WAN3DNS achieves higher accuracy than DeepXDE, NSFnets, and WAN-Biharmonic.

**Strengths:**

- **Novel weak-form adversarial formulation**: This manuscript extends the legacy WAN networks from scalar to vector-valued PDEs..
- **Theoretical guarantees**: This manuscript provides an explicit error bound connecting the network’s residuals to the L2 solution error. The use of Galerkin theory to justify network error behavior is mathematically sound.
- **Experimental rigor**: The benchmarks cover steady, unsteady, and singular flows. The experimental results show clear accuracy gains over prior methods.

**Weaknesses:**

- **Insufficient physical complexity**: The test cases only involve low-Reynolds-number, laminar flows. WAN3DNS’s scalability to turbulent or high-Re regimes remains unclear.
- **No discussions on computational efficiency**: While the adversarial training improves expressivity, it will increase the computational cost and instability, yet runtime and scalability analyses are not thoroughly discussed.
- **Adversarial optimization sensitivity**: The adversarial training process suffers from high sensitivity to hyperparameters (e.g., learning rates, penalty coefficients), which could hinder reproducibility or robustness.
- **No ablation on modern architectures**: Only MLPs are tested. It is unclear how WAN3DNS would perform with more modern architectures (e.g., neural operators, graph neural networks, etc.).

**Questions:**

- How would the proposed WAN3DNS framework scale with domain size and sampling density?
- How are test functions in the adversarial network initialized or regularized? How do the authors ensure numerical stability during the training phase?
- Would replacing the MLPs with operator-learning architectures (e.g., neural operators, graph neural networks) further improve generalization?

---

> ### Author Response · Authors · 2025-11-20
>
> ### Weaknesses:
>
> **Insufficient physical complexity: The test cases only involve low-Reynolds-number, laminar flows. WAN3DNS’s scalability to turbulent or high-Re regimes remains unclear.**
>
> We have added a statement in the revision explicitly positioning this work as a first step toward more challenging turbulent benchmarks, which we plan to include in future extensions.
>
> **No discussions on computational efficiency: While the adversarial training improves expressivity, it will increase the computational cost and instability, yet runtime and scalability analyses are not thoroughly discussed.**
>
> We agree that adversarial training introduces extra cost; we therefore added runtime measurements to all tables and a discussion section analyzing scalability.
>
> **Adversarial optimization sensitivity: The adversarial training process suffers from high sensitivity to hyperparameters (e.g., learning rates, penalty coefficients), which could hinder reproducibility or robustness.**
>
> We have a hyperparameter sweep study (Appendix E) (Already included in the paper submitted) and report robust operating ranges to aid reproducibility.
>
> ### Questions:
>
> **Q1. How would the proposed WAN3DNS framework scale with domain size and sampling density?**
>
> R1. To assess how WAN3DNS scales with respect to the **spatial domain size** and the **sampling density**, we conducted a systematic study on the 3D Beltrami flow benchmark. This will be added in the revision.
> Specifically, we varied:
>
> **Domain size:** (spatial cube, time fixed to \([0,1]\)): $\Omega_1 = [-0.5, 0.5]^3$; $\Omega_2 = [-1, 1]^3$; $\Omega_3 = [-2, 2]^3$
>
> **Number of interior training samples:** $N_{dm}=10,000; 20,000; 30,000$.
>
> **Take-home message:** The sparser the sampling points during training, the larger the error and the lower the accuracy.
>
> | Domain $\Omega$         | $N_{\text{dm}}$ | $\|u - u^{*}\|_{L^2}$ |  $\|v - v^{*}\|_{L^2}$ |  $\|w - w^{*}\|_{L^2}$ |  $\|p - p^{*}\|_{L^2}$ |
> |---------------------------|-------------------|-----------------------------------------|-----------------------------------------|-----------------------------------------|-----------------------------------------|
> | $[-0.5, 0.5]^3$         | 10,000            | 0.016         |  0.019           |      0.018   |     0.013        |
> | $[-0.5, 0.5]^3$         | 20,000            |    0.014    |   0.014    |       0.019  |    0.013   |
> | $[-0.5, 0.5]^3$         | 30,000            |           0.001      |       0.002       |                0.003    |         0.011  |
> | $[-1, 1]^3$             | 10,000            |    0.027           |   0.029     |         0.028   |     0.013   |
> | $[-1, 1]^3$             | 20,000            | 0.010                         |  0.008                          | 0.016                             | 0.011                              |
> | $[-1, 1]^3$             | 30,000            |         0.006             |               0.004         |               0.003      |                  0.009   |
> | $[-2, 2]^3$             | 10,000            |       0.191         |          0.213  |            0.192     |    0.129       |
> | $[-2, 2]^3$             | 20,000            |            0.190       |               0.208        |                0.164     |                 0.103      |
> | $[-2, 2]^3$     | 30,000      |      0.108   |     0.109  |        0.102   |      0.034  |
> -----------------------------
>
> **Q2. How are test functions in the adversarial network initialized or regularized? How do the authors ensure numerical stability during the training phase?**
>
> R2. The adversarial network is initialized or regularized as following:
>
>
>
> 1. All weights of $v$-net are initialized with Xavier-uniform initialization, and all biases are initialized to zero.
>
> 2. To enforce the correct functional structure, we multiply the raw network output by a fixed compactly supported bump function.
>
>
>
> For numerical stability during the training, we use several safeguards:
>
>
>
> 1. Safe gradient computation.
>
> 2. Robust loss design for the adversarial network.
>
> 3. Gradient clipping and handling Non-gradients.
>
> 4. NaN/Inf checks in the loss.
>
> 5. Moderate learning rates and optimizer choice.
>
> 6. Early stopping.
>
>
> We will clarify this in the revised version.
>
>
>
> **Q3. Would replacing the MLPs with operator-learning architectures (e.g., neural operators, graph neural networks) further improve generalization?**
>
> R3. Thank you for this valuable suggestion. Due to the mathematical complexity of weak solutions to the Navier-Stokes equations, we did not extend our MLP-based model directly to operator-learning techniques, which provide a promising direction for improving generalization in weak solution identification. In our future work, we plan to explore operator-learning architectures, such as neural operators and graph neural networks, as a natural extension of the present framework to further enhance generalization; this perspective is now briefly discussed in the revised manuscript.

---

> > ### Author Response · Authors · 2025-11-28
> > **Follow-up Comment**
> >
> > Dear Reviewer,
> >
> > I hope this message finds you well. As the discussion period is concluding, we want to ensure that we have addressed all your concerns. If there is anything else you would like us to clarify, please feel free to let us know.
> >
> > Thank you for your time and effort in reviewing our paper.

---

> ### Author Response · Authors · 2025-12-02
>
> Dear Reviewer,
>
> We are grateful for your careful assessment of the manuscript and for the strengths you pointed out, in particular:
>
> the novelty of extending the weak-form adversarial WAN framework from scalar PDEs to vector-valued incompressible Navier–Stokes systems,
>
> the theoretical guarantees in the form of an explicit error bound linking residuals to the L2 solution error via Galerkin theory,
>
> and the experimental rigor, with benchmarks that cover steady, unsteady, and singular flows where WAN3DNS shows clear accuracy gains over prior methods.
>
> We have now uploaded a revised version of the manuscript and the corresponding rebuttal. In this revision, we respond point by point to all of your comments and questions, and we have added several elements directly motivated by your feedback:
>
> We clarify in the introduction and discussion that the present work focuses on low–Reynolds-number benchmark cases as a first step, and we explicitly position high-Reynolds and turbulent flows as planned future work.
>
> We add runtime measurements to all main tables and provide a short discussion of computational cost and scalability.
>
> We include an additional study on the effect of domain size and sampling density for the 3D Beltrami flow, showing how performance depends on collocation density.
>
> We describe more clearly how the adversarial test functions are initialized and regularized, and we add a figure showing the training dynamics of both the primary and adversarial losses.
>
> We expand the discussion of architectural choices and briefly outline future extensions using operator-learning architectures (e.g., neural operators, graph neural networks).
>
> All new or modified text is highlighted in blue in the revised version to make it easy to identify. Overall, these changes improve clarity, provide stronger experimental evidence, and better illustrate how WAN3DNS can be extended toward more complex and turbulent regimes in future work.
>
> We thank you again for your constructive suggestions and for the positive evaluation of the framework, which have helped us significantly strengthen the manuscript.
>
> Best regards,
>
> Authors

---

### Official Review · Reviewer_cvjj · 2025-11-01

**Soundness:** 1
**Presentation:** 3
**Contribution:** 2
**Rating:** 2
**Confidence:** 4

**Summary:**

The paper introduces WAN3DNS, a deep learning framework for solving the 3D incompressible Navier-Stokes (NS) equations based on the Weak Adversarial Network (WAN) approach. Traditional Physics-Informed Neural Networks (PINNs) rely on the strong formulation of PDEs, requiring high solution regularity and automatic differentiation. In contrast, WAN3DNS leverages the weak (integral) formulation, which relaxes these regularity requirements.

The core methodology recasts the weak form of the NS equations (including the momentum equations and the divergence-free constraint) into a minimax optimization problem. A primary network approximates the velocity and pressure fields, aiming to minimize the residuals in the appropriate dual norms. Simultaneously, an adversarial network generates test functions to maximize these residuals, akin to a GAN training dynamic. This work extends the WAN approach, previously limited to scalar PDEs or 2D NS via stream functions (WAN-Biharmonic), to the full 3D velocity-pressure formulation.

**Strengths:**

The paper presents a well-motivated approach with several significant strengths in the domain of scientific machine learning for fluid dynamics:

1. Originality and Significance: The extension of the WAN framework to the 3D incompressible Navier-Stokes equations in the velocity-pressure formulation is novel and non-trivial. Prior WAN approaches for NS (e.g., WAN-Biharmonic) relied on the 2D stream function formulation, which does not generalize easily to 3D. Successfully handling the coupled vector system and the divergence-free constraint within an adversarial weak-form setting is a valuable methodological contribution.

2. Robustness via Weak Formulation: By leveraging the weak formulation, the method inherently relaxes the differentiability requirements needed by strong-form PINNs. This advantage is clearly demonstrated in the 3D lid-driven cavity (LDC) experiment (Table 3, Figure 9), where WAN3DNS handles the corner singularities much more robustly than the PINN baseline, addressing a known limitation of strong-form approaches.

3. Strong Empirical Performance: The numerical results are compelling. WAN3DNS generally outperforms the SOTA baselines across the benchmarks. The accuracy of the pressure field in the Beltrami flow (Table 2) is particularly impressive, showing orders of magnitude improvement over the baselines (e.g., 0.011 error for WAN3DNS vs. 10.54 for DeepXDE).

4. Theoretical Effort: The inclusion of a theoretical error analysis (Theorem 1) that aims to bound the L2 error by the training residuals is valuable, although the applicability of the proof has critical issues (see Weaknesses).

**Weaknesses:**

Despite the promising results, the paper has significant weaknesses that need to be addressed, particularly concerning the theoretical claims and the rigor of the empirical comparison.

1. Critical Discrepancy Between Theory and Experiments (Major Concern): The theoretical error analysis presented in Appendix B contains a critical flaw regarding its applicability to the numerical experiments. Line 671 explicitly states, "we consider the NS equation with periodic boundary conditions." The definition of the boundary residual also relies on periodicity. However, all numerical experiments (Kovasznay, Beltrami, and LDC) utilize Dirichlet boundary conditions. The proof techniques, particularly the handling of boundary terms arising from integration by parts (e.g., Eqs. 18 and 19), are heavily dependent on the boundary conditions. This mismatch fundamentally undermines the relevance of Theorem 1 to the presented experimental results and severely impacts the soundness of the theoretical claims.

2. Fairness and Rigor of Baseline Comparisons: For the Beltrami flow experiment, the paper states (L375) that the comparison was conducted "using the same hyperparameters" for WAN3DNS, NSFnets, and DeepXDE. This approach is methodologically flawed, as optimal hyperparameters vary significantly across different methods and training strategies (e.g., PINN vs. WAN). A fair comparison requires comparing the best-tuned version of each method.

3. Baseline Specificity: In the 3D LDC experiment (Table 3), the comparison is restricted to a vague "PINNs" baseline, rather than the specific SOTA methods (DeepXDE, NSFnets) used elsewhere. The specific implementation must be detailed for reproducibility.

4. Training Stability and Dynamics: The WAN methodology relies on adversarial training (a minimax game), which is known to be unstable and difficult to tune. While a sensitivity analysis is provided (Appendix E), the paper lacks a discussion on the stability of the training process itself (e.g., convergence plots for both networks, the ratio of training steps, or specific stabilization strategies employed).

5. Limited Scope and High Regularity Assumptions: The experiments are limited to relatively low Reynolds numbers (e.g., Re=400 for LDC). The performance and advantages of WAN3DNS in more challenging, high-Re regimes, where solution regularity is lower, remain unevaluated. Furthermore, the theoretical analysis (Theorem 1) requires strong regularity assumptions on the exact solution, which somewhat contradicts the motivation of using the weak form for low-regularity scenarios.

**Questions:**

I request the authors address the following critical questions during the rebuttal phase:

1. Can the authors address the critical discrepancy regarding the boundary conditions used in the proof of Theorem 1 (periodic BCs mentioned in L672) versus the Dirichlet BCs used in the experiments? Does the proof hold for Dirichlet boundary conditions? If so, please detail how the boundary terms arising from integration by parts (e.g., in Eq. 18 and 20) are rigorously handled or bounded in the non-periodic case.

2. a) For the comparisons in Tables 1 and 2, can you confirm if the baselines (DeepXDE, NSFnets) were independently optimized, rather than just using the hyperparameters tuned for WAN3DNS? If the latter, the comparison is not fair. b) Which specific implementation and architecture were used for the "PINNs" baseline reported in Table 3?

3. In the Beltrami flow (Table 2), the pressure errors for DeepXDE and NSFnets are exceedingly high. Can you elaborate on why the strong-form baselines fail so dramatically in pressure recovery in this specific case, and how the weak adversarial formulation specifically aids in accurate pressure reconstruction?

4. Could the authors provide more insight into the stability of the adversarial training process? A visualization of the loss curves for both the primary and adversarial networks would be helpful. Were specific techniques required to stabilize the minimax optimization?

---

> ### Author Response · Authors · 2025-11-19
>
> **Q1. Can the authors address the critical discrepancy regarding the boundary conditions used in the proof of Theorem 1 (periodic BCs mentioned in L672) versus the Dirichlet BCs used in the experiments? Does the proof hold for Dirichlet boundary conditions? If so, please detail how the boundary terms arising from integration by parts (e.g., in Eq. 18 and 20) are rigorously handled or bounded in the non-periodic case.**
>
> **R1.** Thank you for flagging this.
>
> All experiments in the paper use Dirichlet boundary conditions, and the analysis is carried out in the Dirichlet setting.
>
> Boundary data enter through the boundary mismatch $R_s$ in Eq. (17). After deriving the interior estimate in Eq. (18) and integrating by parts, the boundary integrals appear in Eqs. (19)–(20). These surface terms are not discarded; they are immediately bounded using the boundary residuals $R_{s,u}$ and $R_{s,\nabla u}$, which are precisely penalized by the boundary loss $L_b$ in Eq. (13). The same terms propagate into the final estimate via Eq. (24), where $\|R_s\|_{L^2(\partial \Omega\times [0,T])}$ appears explicitly. Thus, the proof does not rely on periodicity: Dirichlet boundary effects are retained, controlled through $R_s$, and included in the final bound.
>
> In the revision, we will remove **“periodic”** at L 672, state the test spaces explicitly in Sec. 3.1–3.2, and restate Theorem 1 in the Dirichlet form that points back to Eqs. (19)–(21) and Eq. (24).
>
> **Q2. a) For the comparisons in Tables 1 and 2, can you confirm if the baselines (DeepXDE, NSFnets) were independently optimized, rather than just using the hyperparameters tuned for WAN3DNS? If the latter, the comparison is not fair. b) Which specific implementation and architecture were used for the "PINNs" baseline reported in Table 3?**
>
> **R2. a)** We tested multiple hyperparameter settings for all methods. The best performance of each method is reported. Additional results are included in the Appendix, where WAN3DNS continues to show clear advantages.
>
> **b)** The PINNs baseline in Table 3 is implemented in TensorFlow using a standard fully connected MLP with tanh activation, following the classical PINNs formulation. Since existing frameworks do not directly support the lid-driven cavity case, we implemented this model ourselves for a fair comparison.
>
> **Q3. In the Beltrami flow (Table 2), the pressure errors for DeepXDE and NSFnets are exceedingly high. Can you elaborate on why the strong-form baselines fail so dramatically in pressure recovery in this specific case.**
>
> **R3.** Strong-form PINNs struggle with pressure in Beltrami because:
>
> 1. In the strong-form loss, pressure enters only through its gradient, so the optimizer can reduce the momentum residual mainly by adjusting $\mathbf{u}$ while $P$ remains weakly constrained and sensitive to high-order automatic differentiation.
>
> 2. High-order Automatic Differentiation on $\Delta \mathbf{u}$ and $\nabla P$ causes noise amplification and instability in pressure.
>
> **Q4. How the weak adversarial formulation specifically aids in accurate pressure reconstruction?**
>
> **R4.** By incorporating the pressure term only in the weak form and enforcing incompressibility through adversarial test functions, our formulation provides the following advantages:
>
> 1. Pressure appears only variational coupling $(P, \nabla \cdot v)$ with learned test functions, avoiding pointwise instability.
>
> 2. The divergence-free constraint is enforced globally, ensuring physically consistent pressure–velocity coupling.
>
> 3. Only lower-order derivatives are involved in the pressure reconstruction, reducing numerical sensitivity and regularity requirements.
>
> **Q5. Could the authors provide more insight into the stability of the adversarial training process? A visualization of the loss curves for both the primary and adversarial networks would be helpful. Were specific techniques required to stabilize the minimax optimization?**
>
> **R5.** We thank the reviewer for this helpful suggestion. In the revised manuscript, we have included the visualization of the training dynamics, showing both the primary loss and the adversarial loss.
>
> The adversarial loss exhibits fluctuations, which is theoretically expected: the adversarial network represents arbitrary test functions used to maximize the weak constraint violation. Since the maximization objective continuously adapts to challenge the current solution, the adversarial network does not necessarily converge to a fixed state, leading to non-monotonic dynamics.
>
> Importantly, these fluctuations do not indicate instability: they are a natural and desirable characteristic of minimax optimization, analogous to saddle-point dynamics in mixed variational formulations.
>
> To ensure robust training in practice, we also applied commonly used stabilization techniques, including asymmetric learning rates, gradient clipping, and early stopping once the adversarial loss variations indicate sufficiently enforced incompressibility.

---

> > ### Author Response · Authors · 2025-11-28
> > **Follow-up Comment**
> >
> > Dear Reviewer,
> >
> > I hope this message finds you well. As the discussion period is concluding, we want to ensure that we have addressed all your concerns. If there is anything else you would like us to clarify, please feel free to let us know.
> >
> > Thank you for your time and effort in reviewing our paper.

---

> > > ### Comment · Reviewer_cvjj · 2025-11-28
> > >
> > > Dear Author(s),
> > >
> > > Please submit the revised article with red/blue labels for the updates such that I can review based on both the comments and the updated article. As of now I can only see the original version (updated 10.8)

---

> ### Author Response · Authors · 2025-12-01
>
> Dear Reviewer,
>
> Thank you very much for your thorough review and for the strengths you highlighted. We particularly appreciate that you emphasized:
>
> the originality and significance of extending the WAN framework to the full 3D incompressible Navier–Stokes equations in the velocity–pressure formulation,
>
> the robustness benefits of the weak formulation, especially in the 3D lid-driven cavity case where WAN3DNS handles corner singularities better than strong-form PINNs,
>
> the strong empirical performance of WAN3DNS across the benchmarks, including the significant improvement in pressure reconstruction for Beltrami flow,
>
> and the value of including a theoretical error analysis that links the training residual to an L2-type error, even though you rightly pointed out limitations in its original applicability.
>
> We have now uploaded a revised version of the manuscript and an updated rebuttal. In this revision, we address your comments and questions point by point, with several key changes:
>
> We clarify that Theorem 1 is formulated and proved under Dirichlet boundary conditions, and we make explicit how the boundary residuals and boundary terms from integration by parts are controlled in the analysis.
>
> We correct and streamline the proof in the appendix and make the link between the residuals and the final error bound more transparent.
>
> We explain more clearly the experimental setup and the fairness of baseline comparisons, and we describe the PINNs baseline used for the lid-driven cavity case.
>
> We elaborate on why strong-form methods struggle with pressure in the Beltrami flow and how the weak adversarial formulation helps stabilize pressure recovery.
>
> We add a figure showing training dynamics for both the primary and adversarial networks, and we describe the stabilization techniques used in practice (learning-rate asymmetry, gradient clipping, early stopping, etc.).
>
> All new or modified text is highlighted in blue in the revised version to facilitate your review. Overall, these changes strengthen the connection between the theoretical analysis and the experimental setting, clarify the role and limitations of Theorem 1, and better justify the advantages of the weak adversarial framework in 3D velocity–pressure Navier–Stokes problems.
>
> We are very grateful for your careful and critical feedback, which has significantly improved both the rigor and the clarity of the paper.
>
> Best regards,
>
> Authors

---

### Official Review · Reviewer_7Bz7 · 2025-11-01

**Soundness:** 1
**Presentation:** 1
**Contribution:** 1
**Rating:** 2
**Confidence:** 4

**Summary:**

This paper presents a neural network method to solve the incompressible Navier-Stokes equations (NSE). The specific method is a weak adversarial approach, in which the weak form of the equations is considered, and a min-max problem is formulated: minimization or the residuals with respect to the trial functions, and maximization of the residuals with respect to the test functions. This method had been introduced in Zhang et al (2020) and is cited in this manuscript. This work extends the method of Zhang et al.to vector valued problems (in particular, NSE). The authors present a brief theoretical analysis of the errors, and test their method on three well known benchmark problems, and compare against WAN, DeepXDE and NSFNets methods.

**Strengths:**

* The strengths of this method derives from the strength of the parent method (Zhang et al (2020)). Zhang et al (2020)'s paper presents an insightful mathematical formulation of the PDE weak formulation into a min-max problem and using a network and an adversarial training algorithm. The current paper applies that method to the incompressible Navier-Stokes equations.

* This paper is readable. But there are weaknesses (see below).

* And the authors have presented results on various examples of the Navier-Stokes equations.

* A theoretical analysis on the boundedness of the error is presented.

* Comparison with a few other methods is provided.

**Weaknesses:**

* No new insight.

* Some notations are not properly introduced. It is possible that the paper assumes that the reader is familiar with the original method (Zhang et al. (2020)) and the notation therein. In that case, this must be explicitly stated.

* This paper is readable. But the writing needs to be improved. The mathematical statements (such as Theorem 1) need to be precise, all symbols must be explained, and must have their role to play

* The authors presented a theorem on the bounds of the errors, but did not show any numerical result relating to the estimates, at least not in the main text.


$\textbf{Suggestions}$:

* Line 210: If $\mathbf{u}$ is a vector valued function, it is recommended to use the notation $\mathbf{u}:[0,T]\to H^1_0(\Omega, \mathbb{R}^m)$ or $\mathbf{u}:[0,T]\to [H^1_0(\Omega)]^m$.

* Line 217: It would be better to explicitly state $d=2$ or 3.

* Line 228: Should be $u_i$ on the RHS.

**Questions:**

* Eq 10 (and in many subsequent instances): $\mathcal{A}_t[u]$ is now only a function of $u$, and not of $[u,p]$, why?

* Line 245: is $p$ included in $u_\theta$?

* Theorem 1: what is $q_\theta,\ R_s,\ R_{PDE}$.. ?

* Theorem 1: I am assuming that the theorem must have a constraint on the viscosity parameter. Is that correct? And did the authors consider this?

* Theorem 1: In the case that the authors assumed a high viscosity regime, did the authors invoke the classical inf-sup condition of the weak form? And did the authors attempt to obtain a similar error estimate on the pressure?

---

> ### Author Response · Authors · 2025-11-20
>
> We respectfully disagree with the statement that our work presents “no new insight.” While the 3D incompressible Navier–Stokes equations remain a Clay Millennium Prize Problem with many theoretical aspects still open, applying learning-based methods to this system via a variational adversarial framework is nontrivial and requires substantial reformulation. In particular, we extend a weak adversarial formulation to 3D vector-valued fields with divergence-free constraints and pressure coupling, which introduces genuinely new mathematical and computational challenges. Moreover, in many realistic 3D flow configurations strong solutions are not available, so having a simple and practical algorithm that targets weak solutions is of independent importance.
>
> In the revision: We will (i) supplement a clear notation table to ensure full readability, (ii) rewrite Theorem 1 to make the statement fully precise, and (iii) include a detailed proof and a clarifying remark.
>
> In Theorem 1 the error bound depends on Poincaré-type constants and Gronwall’s inequality that cannot be computed explicitly in practical neural network implementations. For this reason, the theoretical result is not used directly as an error indicator during training.
>
> **Q1: Eq 10 (and in many subsequent instances): $\mathcal{A}_t[\mathbf{u}]$ is now only a function of $\mathbf{u}$, and not of $[\mathbf{u}, P]$, why?**
>
> R1: This is a typo. In Eq. (10) and subsequent occurrences, the operator should in fact be written as $\mathcal{A}_t[\mathbf{u}, P]$ rather than only $\mathcal{A}_t[\mathbf{u}]$. The pressure $P$ is part of the argument of the operator and enters the formulation together with the velocity field $\mathbf{u}$.
>
> **Q2: Line 245: is $P$ included in $\mathbf{u}_\theta$?**
>
> R2: $P$ is indeed included in the neural network parametrization: the vector $\theta$ parametrizes both the velocity $\mathbf{u}_{\theta}$ **and** the pressure $P_{\theta}$. We will correct the notation to make this dependence explicit and avoid confusion.
>
> **Q3: Theorem 1: what is $q_\theta$, $R_s$, $R_{\text{PDE}}$.. ?**
>
> R3: There is no $q_{\theta}$, this is a typo;
>
> $R_{s}=(R_{s,u},R_{s,P},R_{s,\nabla u})$ is the residuals related to the periodic boundary conditiona, where
>
> $R_{s,u}(x)=u_{\theta}(x)-u_{\theta}(x+\xi)$, $R_{s,P}(x)=P_{\theta}(x)-P_{\theta}(x+\xi)$,
>
> $R_{s,\nabla u}(x)=\nabla u_{\theta}(x)-\nabla u_{\theta}(x+\xi)$, $\xi$ is the period.
>
> $R_{PDE}=\partial_{t}u_{\theta}+(u_{\theta}\cdot \nabla)u_{\theta}+\nabla P_{\theta}-\nu\Delta u_{\theta}$ is the residuals related to the governing equation.
>
> **Q4: Theorem 1: I am assuming that the theorem must have a constraint on the viscosity parameter. Is that correct? And did the authors consider this?**
>
> R4: There is no constraint on the viscosity parameter. In all basic setup, the viscosity is denoted by $\nu$, it should be $>0$. No other constraint on it.
>
> **Q5: Theorem 1: In the case that the authors assumed a high viscosity regime, did the authors invoke the classical inf-sup condition of the weak form?**
>
>
> R5: We clarify:
>
>
>
> 1. *No assumption of high viscosity.* We clarify that our method does not rely on or assume a high viscosity regime.
>
>
> 2. We did not explicitly invoke the inf-sup condition in the original manuscript.
>
> 3. The inf-sup condition ensures pressure stability through the divergence constraint, and is not related to the viscosity coefficient.
>
>
>
> 4. *Definition of the inf-sup condition.* For the mixed velocity-pressure weak formulation with $V=H^{1}(\Omega)^{d}$ (velocity space) and $P=L^{2}(\Omega)$ (pressure space), the classical inf-sup condition is
>
> $\inf_{q\in Q}\sup_{v\in V}$ $=\frac{-(\nabla \cdot v,q)}{\|v\|\cdot \|q\|}\ge\beta >0$
>
> Here, $v$ is velocity and $q$ is pressure. This condition guarantees well-posedness and pressure uniqueness in compressible flow problems.
>
> 5. *The inf-sup condition is automatically satisfied in our framework.* The weak divergence condition is enforced through the maximization of dual test functions within the adversarial framework. This mechanism preserves the essential role of the inf-sup condition without requiring an explicit discrete verification. Therefore, our formulation inherits the continuous inf-sup stability guaranteed by the underlying functional spaces.
>
> **Q6: And did the authors attempt to obtain a similar error estimate on the pressure?**
>
> R6: We do not provide a similar error estimate for the pressure. Indeed, the focus of this paper is on the estimation of an error bound for the velocity field. But we can provide an idea about the pressure estimate.  We would like to note that one can also prove a stability result for $\|P-P_{\theta}\|_{L^{2}(\Omega)}$ in a similar spirit to Theorem 1. The main steps consist of taking the divergence of the Navier-Stokes equations, using the identity
>
> $-\Delta P=Trace ((\nabla u)^{2})=\sum_{i,j}u_{x_{j}}^{i}u_{x_{i}}^{j}$
>
> and rewriting the result in terms of the different residuals.

---

> > ### Author Response · Authors · 2025-11-28
> > **Follow-up Comment**
> >
> > Dear Reviewer,
> >
> > I hope this message finds you well. As the discussion period is concluding, we want to ensure that we have addressed all your concerns. If there is anything else you would like us to clarify, please feel free to let us know.
> >
> > Thank you for your time and effort in reviewing our paper.

---

> ### Author Response · Authors · 2025-12-02
>
> Dear Reviewer,
>
> We would like to sincerely thank you for your careful reading of our manuscript and for highlighting several strengths of the work, in particular:
>
> the value of the original WAN formulation of Zhang et al. (2020) and the extension of that min–max weak-form idea to the incompressible Navier–Stokes equations,
>
> the fact that the paper is readable and presents results on several Navier–Stokes benchmarks,
>
> the inclusion of a theoretical analysis on the boundedness of the error,
>
> and the comparison with several existing methods.
>
> We have now uploaded a revised version of the manuscript and the corresponding rebuttal. In this revision, we respond point by point to all of your comments and questions, and we have made several concrete changes to improve the paper:
>
> We clarify the notation throughout the paper and add a notation table to improve readability.
>
> We rewrite Theorem 1 to make the statement and all symbols fully precise, and we correct and streamline the proof in the appendix.
>
> We specify more clearly how the viscosity parameter and the weak formulation enter the analysis, and we clarify the role of the divergence constraint and stability (inf–sup perspective).
>
> We improve the explanations around the error bound and its practical interpretation.
>
> All new or modified text is highlighted in blue in the revised version to make it easy to identify. Overall, these revisions improve clarity, provide stronger theoretical grounding, and better explain the contribution of extending the WAN framework to 3D velocity–pressure Navier–Stokes systems.
>
> We thank you again for your time, for your detailed comments, and for the constructive feedback that helped us make the manuscript more precise and accessible to readers.
>
> Best regards,
>
> Authors

---

### Meta-Review · Area_Chair_jrjA · 2026-01-06

**Summary:**

The reviewers generally indicate that:
1. The work is an extension of the WAN formulation of Zhang et al. (2020) to the 3D incompressible Navier–Stokes equations which is non-trivial.
2. Baseline comparisons are not done with the optimal hyper parameters for the respective method.
3. Performance of the proposed method achieves significant improvements over the baselines.
4. The assumptions of the presented theory are not satisfied by the numerical experiments.
5. The numerical experiments are limited to laminer flows with low Reynolds numbers and are on simple domains (box).
6. Optimization of minimax problems in practice is usually unstable and the authors provide few details on how they deal with this issue.

**Reviewer Concerns:**

The authors have addressed most reviewer concerns well. In particular, they have clarified the their choices for different baselines and their hyperparmeters, modified the theory to match the assumptions of their numerical experiments, and added details on how to practically solve the minimax problem. The biggest issue which remains is that the chosen numerical examples are quite simple. For a box domain and a low Reynolds number, the 3D NS equations can be solved relatively efficiently with classical solver. The paper provides no comparisons of the speed of their method to classical method but only to other PINNs. This is quite important in this case as it would showcase why their method is useful for this particular problem rather than being just another variation on a PINN.

**Reviewer Scores:**

Reviewers 7Bz7 and cvjj may have raised their scores to 4 given the extensive rebuttal provided by the authors. However the paper is still missing a fundamental comparison needed to make a strong argument for their method.

---

### Decision · Program_Chairs · 2026-01-26

Reject